# Regulating cleavage activity and enabling microRNA detection with split sgRNA in Cas12b

Jiaqi Wang[1,4], Xiaofang Ye[1,4], Yuanfang Liu[1,4], Wentao Li[2], Xue Zhang[1], Wei Zhang[1], Changqing Yi [3] & Chaoxing Liu [1] ✉

The CRISPR-Cas12b system has revolutionized molecular diagnostics, yet its reliance on single guide RNAs (sgRNAs) exceeding 100 nt limits precise regulation and applications. We present a split sgRNA strategy for Cas12b, utilizing universal components with customizable Spacer to detect various nucleic acid targets by simply replacing Spacer. Glyoxal labeling of the universal split direct repeat (DR) region represses Cas12b activity, which is restored by elevated temperatures or prolonged incubation, enabling dynamic regulation. Incorporating a photo-cleavable linker into the DR allows UV-mediated modulation, ensuring compatibility with recombinase polymerase amplification. Successful detection of Epstein-Barr virus in clinical plasma samples matched the sensitivity of traditional qPCR. Importantly, microRNAs can replace the Spacer, enabling direct detection without reverse transcription or amplification. Supported by evidence from cultured cells and plasma from healthy individuals and colorectal cancer patients, this method yields consistent results with RT-qPCR while simplifying protocols. This split strategy enhances Cas12b systems, offering a promising approach for clinical nucleic acid analysis.

Cas12b, previously known as C2c1, is a class 2 type V-B endonuclease. It consists of an NUC lobe and a REC lobe, but lacks an HNH domain[1]. Cas12b exhibits both *cis*- and *trans*-cleavage activities[2,3] towards specific nucleic acids by utilizing single guide RNA (sgRNA) or CRISPR RNA (crRNA) with *trans*-activating CRISPR RNA (tracrRNA)[4–6]. Compared to Cas9 and Cas12a, Cas12b is smaller, making it more suitable for gene therapy applications via adeno-associated virus delivery[7,8]. Its optimal cleavage activity spans a broad temperature range of 31 °C to 67 °C[7,9], with detectable activity even from 4 °C to 100 °C[7]. Furthermore, Cas12b demonstrates robust cleavage efficacy across a wide pH range of 1.0 to 8.0, and it can withstand conditions ranging from pH 1.0 to 12.0[7]. Remarkably, even after being incubated with human plasma at 37 °C for 12 hours, Cas12b retains significant activity and effectively cleaves DNA targets[7]. These advantageous characteristics position Cas12b as a key player in gene editing[7,8,10–12] and in the detection of nucleic acid mutations with single-base resolution[3], particularly for clinical pathogens such as SARS-CoV-2[13,14], monkeypox virus (MPXV)[15], Mycobacterium tuberculosis[16], and hepatitis C virus (HCV)[9].

However, there are still areas in which Cas12b could be improved in comparison to Cas12a, which requires only approximately 42 nucleotides (nt) of crRNA for effective function[17–20], while Cas12b necessitates around 111 nt of sgRNA[21]. The synthesis of RNA greater than 100 nt through nucleic acid solid-phase synthesis poses challenges and incurs high costs[22]. Currently, the preferred method involves transcribing DNA into RNA using T7 RNA polymerase, but this method presents challenges in the chemical modification of the sgRNA

[1]Guangdong Provincial Key Laboratory of Digestive Cancer Research, Digestive Diseases Center, Scientific Research Center, The Seventh Affiliated Hospital of Sun Yat-sen University, Shenzhen, Guangdong, China. [2]Department of Clinical Laboratory, The Seventh Affiliated Hospital of Sun Yat-sen University, Shenzhen, Guangdong, China. [3]Guangdong Provincial Key Laboratory of Sensor Technology and Biomedical Instrument, School of Biomedical Engineering, Shenzhen Campus of Sun Yat-Sen University, Shenzhen, Guangdong, China. [4]These authors contributed equally: Jiaqi Wang, Xiaofang Ye, Yuanfang Liu. ✉e-mail: liuchx69@mail.sysu.edu.cn

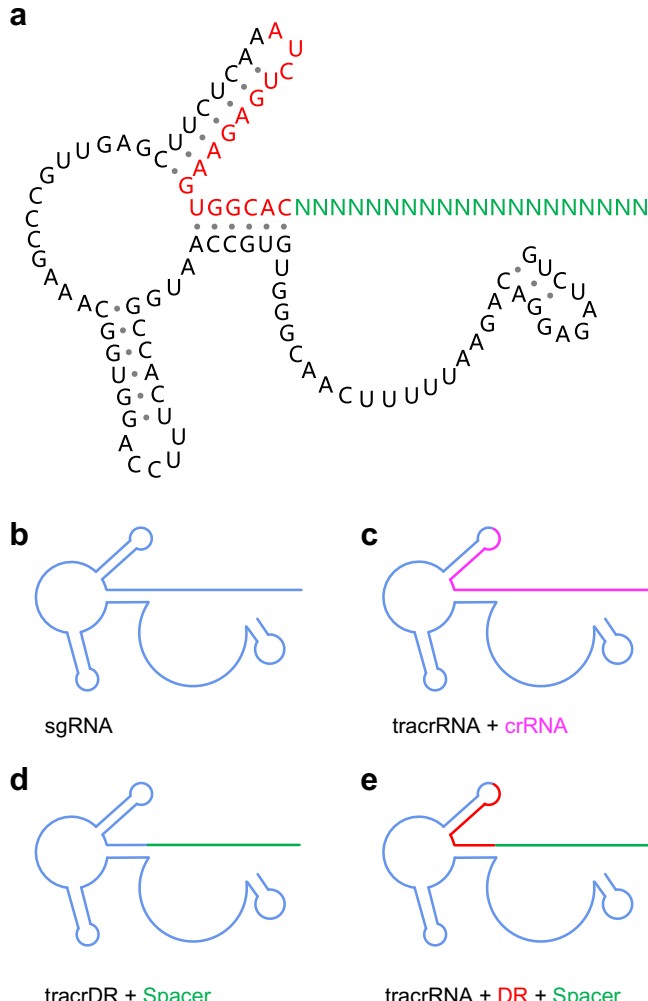

**Fig. 1 | Overview of sgRNA and split sgRNA strategies for Cas12b. a** The complete sequence and potential secondary structure of sgRNA for Cas12b. **b** Illustration of the full-length sgRNA designed for Cas12b functionality. **c** Overview of the traditional split strategy, incorporating tracrRNA and crRNA for Cas12b, where the blue segment denotes tracrRNA and the pink segment represents crRNA. Our currently developed split sgRNA strategies for Cas12b are illustrated as follows: **d** tracrDR+Spacer and **e** tracrRNA+DR+Spacer. The red region indicates the universal DR region (16 nt), while the green region depicts the interchangeable Spacer region (18–23 nt).

produced[23]. Nevertheless, synthesizing sgRNA with modified nucleobases presents an effective solution to address the inherent incompatibilities of isothermal amplification and CRISPR-based one-pot nucleic acid detection methods[24–28]. In scenarios where low amounts of nucleic acids are subjected to isothermal amplification, concurrent CRISPR reactions may deplete the amplified templates, limiting detection sensitivity at very low sample concentrations[24–26]. Therefore, a crucial focus in this field is to enhance Cas12b's performance while maintaining its advantages, and to extend its applications while mitigating its limitations.

Herein, we split Cas12b sgRNA into two components: tracrDR and Spacer (Fig. 1d), or into three parts: tracrRNA, DR, and Spacer (Fig. 1e). We hypothesize that this split sgRNA will still guide the cleavage of Cas12b. Importantly, the universal nature of tracrDR or tracrRNA+DR allows us to easily replace only Spacer (~18–23 nt) to detect different targets. In addition, the short universal DR moiety (11 nt) is easily artificially synthesized and chemically modified, allowing precise regulation of Cas12b activity by different modifications and de-modifications of DR. In particular, the length of the split Spacer is similar to that of microRNA[29], which opens the possibility of

replacing the original Spacer with microRNA to guide the cleavage activity of Cas12b. This feature also opens up avenues for the development of a CRISPR12b-based detection system that specifically targets microRNA. While preparing this manuscript, a similar study using Cas12a split crRNA for microRNA detection was published[30], which further emphasizes the importance of our systematic investigation of Cas12b split sgRNA, along with its promising potential for future development, including several promising applications recently identified in the Cas12a split crRNA strategy[30–33].

## Results

### Exploration of cleavage activity in the split sgRNA-assisted CRISPR-Cas12b system

To investigate whether the split sgRNA can bind to Cas12b and effectively guide the *cis*-cleavage of target nucleic acids as well as the *trans*-cleavage of single-stranded random nucleic acids, we designed and synthesized targeting EBV full-length sgRNA, classical split structures (tracrRNA+crRNA), and our primary focus—the more universal split form consisting of two components: the tracrDR region and a replaceable Spacer region (tracrDR+Spacer), or three components: tracrRNA, DR, and a replaceable Spacer (tracrRNA+DR+Spacer) (Fig. 1).

We then prepared a 357 bp double-stranded EBV target and a FAM-labeled 15 nt random single-stranded nucleic acid (FAM-ssDNA, Supplementary Table 1). In the presence of the double-stranded target, the full-length sgRNA was able to assemble with Cas12b, thereby activating its *trans*-cleavage activity to degrade FAM-ssDNA. As predicted, denaturing PAGE analysis revealed that our split form, tracrDR+Spacer and tracrRNA+DR+Spacer, demonstrated *trans*-cleavage activation of Cas12b comparable to that of full-length sgRNA and classical tracrRNA +crRNA (Fig. 2a).

Subsequently, we replaced the double-stranded EBV target with a single-stranded EBV target and observed that the split sgRNA retained its ability to assist in the activation of Cas12b's *trans*-cleavage (Supplementary Fig. 1). We further examined the system using an ssDNA reporter with a fluorophore FAM at one end and a quencher BHQ1 at the other end (F-Q, Supplementary Table 1). In solution, the proximity of the quencher to the fluorophore resulted in fluorescence quenching. However, following the activation of *trans*-cleavage by Cas12b, F-Q was degraded, leading to the separation of the quencher from the fluorophore and the subsequent release of a detectable fluorescence signal. Our experimental data demonstrated that the split structures, tracrDR+Spacer and tracrRNA+DR+Spacer, effectively activated Cas12b's *trans*-cleavage activity, resulting in the emission of fluorescence signals akin to those produced by full-length sgRNA and classical tracrRNA+crRNA (Fig. 2b). We then quantitatively analyzed the *trans*-cleavage activity of Cas12b assisted by both full-length and split sgRNA. In all cases, the reaction velocity increased with substrate concentration (Fig. 2c). When engaging target DNA with full-length sgRNA, the catalytic efficiency ($k_{cat}/K_M$) was measured at $7.69 \times 10^4\,M^{-1}\,S^{-1}$. The catalytic efficiency slightly decreased to $4.98 \times 10^4\,M^{-1}\,S^{-1}$ when using the classical tracrRNA+crRNA, indicating robust cleavage activity. Excitingly, our split structures exhibited similar catalytic efficiencies, with tracrDR+Spacer at $4.29 \times 10^4\,M^{-1}\,S^{-1}$ and tracrRNA+DR+Spacer at $4.12 \times 10^4\,M^{-1}\,S^{-1}$. Although there was a marginal decrease in *trans*-cleavage activity compared to full-length sgRNA, it remained highly effective (Fig. 2d, Supplementary Fig. 2).

Next, we systematically investigated the *cis*-cleavage ability of both full-length and split sgRNA. Our results showed that both forms possessed effective *cis*-cleavage abilities. With consistent concentrations of Cas12b and target nucleic acids, an increase in the concentration of either full-length or split sgRNA, or along with extended incubation time, resulted in a proportional increase in the bands corresponding to *cis*-cleavage products (Supplementary Fig. 3). However, we observed that under identical concentrations and time, the split

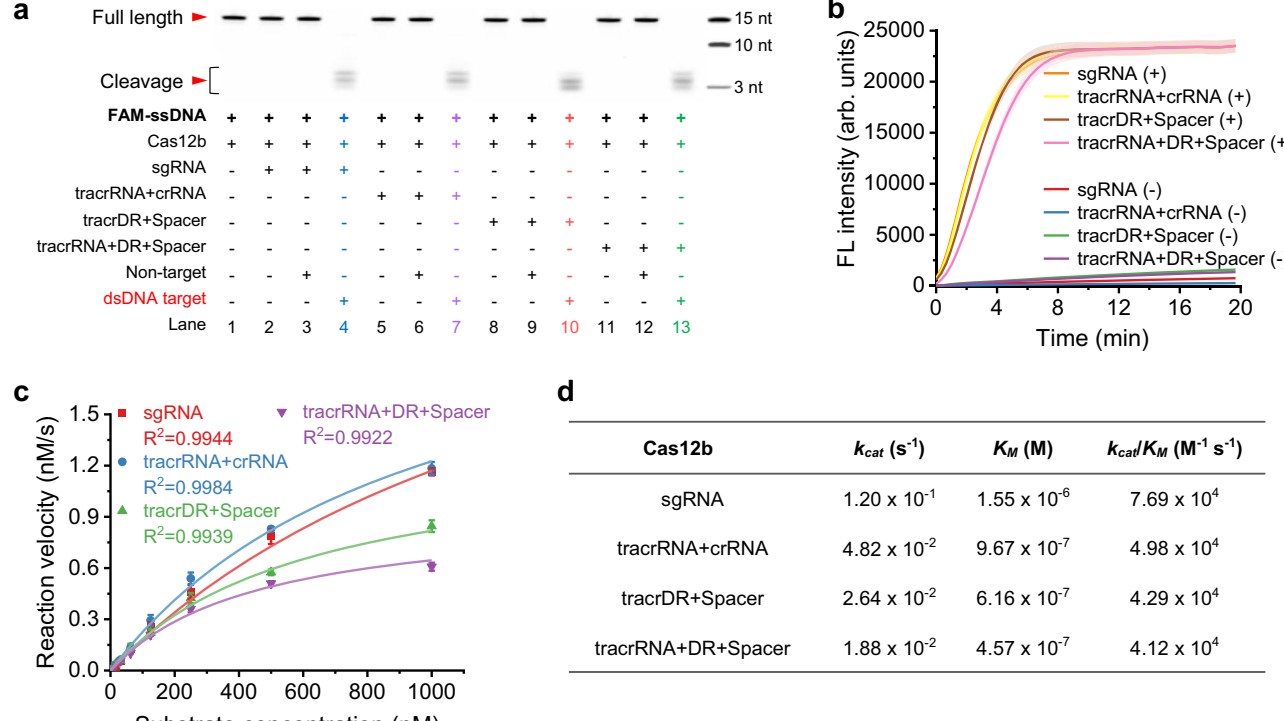

**Fig. 2 | Functional assessment of split sgRNA demonstrating equivalent *trans*-cleavage activity to full-length sgRNA. a** Denaturing PAGE analysis demonstrates the effective *trans*-cleavage activity of tracrDR+Spacer and tracrRNA+DR+Spacer, comparable to that of full-length sgRNA and tracrRNA+crRNA. FAM-ssDNA denotes a FAM-labeled single-stranded random DNA strand that is cleaved upon activation of the *trans*-cleavage activity of the Cas12b system. dsDNA target refers to the double-stranded target DNA that is targeted by the sgRNA, whereas Non-target indicates double-stranded DNA that is not recognized by sgRNA. The experiment was performed three times. **b** Fluorescent signal measurements provide supporting evidence for the *trans*-cleavage activity for both tracrDR+Spacer and tracrRNA+DR+Spacer, with results comparable to those observed for full-length sgRNA and tracrRNA+crRNA. This system includes a reporter (F-Q) whose fluorescence is released following the digestion action of Cas12b. (+) signifies the presence of the optimal double-stranded target DNA, while (−) denotes double-stranded DNA that is not targeted by the sgRNA. Data are shown as mean value +/− SD ($n = 3$, biologically independent samples). **c** Michaelis–Menten kinetic curves illustrate the relationship between Cas12b reaction velocity and substrate concentration when using either full-length or split sgRNA. All the experiments were conducted in triplicate and error bars represent mean value +/− SD ($n = 3$). **d** Comparative analysis of the Michaelis–Menten kinetic parameters for Cas12b employing either full-length or split sgRNA. Source data are provided as a Source Data file.

structures, tracrDR+Spacer and tracrRNA+DR+Spacer, produced fewer *cis*-cleavage products compared to full-length sgRNA and classical tracrRNA+crRNA (Supplementary Figs. 3, 4). This may present an advantage in applications involving one-pot isothermal amplification-CRISPR detection, where the maintenance of similar levels of *trans*-cleavage activity (Fig. 2b, d) while producing lower amounts of *cis*-cleavage products (Supplementary Figs. 3, 4) could reduce template cleavage during trace-level template amplification. In contrast, for applications requiring higher yields of *cis*-cleavage products, increasing the concentration of the split sgRNA or extending the reaction time may increase the yield of *cis*-cleavage products.

## Mechanistic insights into split sgRNA strategies on Cas12b binding and cleavage activity

The activation of Cas12b involves the formation of a ribonucleoprotein (RNP) complex by the binding of sgRNA to Cas12b, followed by the interaction of the RNP with the target (activator)[6]. To investigate how different split sgRNA strategies affect the binding of sgRNA to Cas12b and the target, and their subsequent effect on cleavage activity, we performed a series of in-depth experiments. First, we used microscale thermophoresis (MST) assays to determine the dissociation constants (Kd) of full-length sgRNA and various split sgRNA forms with Cas12b. Our results showed that full-length sgRNA exhibited the lowest Kd (241 ± 11 nM), indicating the strongest binding affinity. In contrast, the classical split form (tracrRNA+crRNA) showed a slightly higher Kd (310 ± 32 nM), while our split forms, tracrDR+Spacer (641 ± 61 nM) and

tracrRNA+DR+Spacer (1070 ± 100 nM), showed significantly higher Kd values, indicating reduced binding affinity compared to full-length sgRNA (Supplementary Fig. 5).

Next, we measured the dissociation constants of the RNP complexes formed by full-length and split sgRNAs with target DNA. Full-length sgRNA and the classic split form (tracrRNA+crRNA) exhibited the lowest Kd values (1.1 ± 0.1 μM and 1.0 ± 0.1 μM, respectively). In contrast, our split forms, tracrDR+Spacer (4.1 ± 0.7 μM) and tracrRNA+DR+Spacer (6.4 ± 0.9 μM), showed significantly higher Kd values, indicating weaker interactions with the target DNA (Supplementary Fig. 6). Notably, Cas12b's *cis*- and *trans*-cleavage activities share the same catalytic site[6]. Previous studies on Cas12a have shown that *trans*-cleavage initiates only after the dissociation of *cis*-cleavage products[34]. Our gel experiments revealed that while our split strategies exhibited similar *trans*-cleavage activity, they produced fewer *cis*-cleavage products (Supplementary Fig. 4). This could be attributed to the higher dissociation constants of our split forms, leading to faster target dissociation and enhanced *trans*-cleavage activity when *cis*-cleavage products are comparable.

To further elucidate the structural dynamics, we employed AlphaFold3[35] to predict the 3D structures of Cas12b complexes with full-length and three split sgRNAs. Molecular dynamics (MD) simulations were performed using GROMACS 2024.5 with the amber14s-b_OL15 force field for 100 ns[36]. The root mean square deviation (RMSD) analysis indicated that the systems stabilized after 30 ns, suggesting the simulations reached a stable conformational state

(Supplementary Fig. 7a). The average binding free energy analysis[37] revealed that full-length sgRNA had the lowest binding free energy, aligning with the MST results (Supplementary Fig. 7b). Root mean square fluctuation (RMSF) analysis was conducted to assess the stability of Cas12b in the four systems. The REC1 region (including residues 100-200) exhibited significant conformational changes (Supplementary Fig. 7c). Specifically, the Cas12b-sgRNA complex REC1 region achieved stability within 30 ns, while the Cas12b-tracrRNA+crRNA complex stabilized after 90 ns. The Cas12b-tracrDR+Spacer complex reached stability at 60 ns, whereas the Cas12b-tracrRNA+DR+Spacer complex showed continuous conformational fluctuations throughout the 100 ns simulation (Supplementary Fig. 8). These results suggest that full-length sgRNA forms a more stable REC1 region of the complex with Cas12b compared to split forms.

The early stabilization of the REC1 region in the Cas12b-sgRNA complex (within 30 ns) facilitates the formation of an optimal target DNA binding pocket, accelerating the specificity recognition process[5,6,34]. This aligns with the enzyme-substrate induced fit theory[38] and is consistent with the MST results showing the lowest Kd for the interaction between full-length sgRNA and Cas12b. Cas12b utilizes a single RuvC nuclease domain for both *cis*-cleavage of target dsDNA and *trans*-cleavage of non-specific ssDNA. We analyzed the Gibbs free energy contributions of the RuvC domain in the four systems. The tracrDR+Spacer split form exhibited Gibbs free energy contributions similar to full-length sgRNA, while the tracrRNA+DR+Spacer form interacted with more RuvC residues but with weaker interactions (Supplementary Fig. 9). Electrostatic and hydrogen bond analyses of the RuvC and Spacer regions revealed that the interactions between full-length and split sgRNAs with Cas12b involved different amino acid residues (Supplementary Fig. 10). This indicates that different split strategies may lead to distinct structural changes in the interactions between sgRNA and Cas12b's functional amino acids. Our findings demonstrate that full-length sgRNA forms the most stable complex with Cas12b, exhibiting the highest binding affinity and optimal cleavage activity. The split forms, while showing reduced binding affinity, reduced *cis*-cleavage activity, still maintain significant *trans*-cleavage activity. These findings contribute to a deeper understanding of how split sgRNAs affect Cas12b cleavage activity and potential applications in CRISPR-Cas12b-based technologies.

## Universal strategy for detecting various nucleic acid targets by simply replacing only the matching Spacer region

To assess the viability of using a universal tracrDR or tracrRNA+DR system to match different 18–20 nt Spacer regions for detecting different nucleic acid targets, we designed and synthesized various Spacer regions targeting EBV, MPXV, and HCV (Supplementary Table 2). Our results indicated that employing a universal tracrDR and matching different Spacer regions proved highly effective for detection various nucleic acid targets. The limits of detection (LODs) for EBV, MPXV, and HCV were 3.18, 3.09, and 12.4 pM, respectively, with good linearity. (Fig. 3a, b, e, Supplementary Fig. 11a, 11b, 11e, 11f). Likewise, coupling the tracrRNA+DR system with different Spacer regions also facilitated effective detection of the aforementioned viral samples. The LODs for EBV, MPXV, and HCV were 15.9, 5.35, and 16.4 pM, respectively (Fig. 3c–e, Supplementary Fig. 11c, d, g, h). Comparatively, the tracrRNA+DR+Spacer strategy exhibited slightly poorer LODs than the tracrDR+Spacer strategy, and the difference is not really significant. Furthermore, the detection performance of the tracrDR or tracrRNA+DR approach with EBV Spacer was comparable to that of full-length sgRNA and tracrRNA+crRNA, without significant differences in magnitude (Supplementary Fig. 12, LODs for sgRNA, tracrRNA +crRNA, tracrDR+Spacer, tracrRNA+DR+Spacer were 0.556, 0.830, 3.18, 15.9 pM, respectively). Additionally, our method showed promising potential for early detection of colon cancer (Supplementary

Fig. 13). The Septin9 gene's high methylation levels serve as a biomarker for the early detection of colon cancer[39]. We prepared simulated samples by varying the proportions of highly methylated Septin9 artificial synthetic DNA sequences (Supplementary Table 2). These nucleic acid samples were pre-treated using the TAPS method, where 5-methylcytosine (5mC) was selectively converted to dihydrouridine (DHU), an analog of T[40]. Subsequently, the processed samples underwent RPA, causing the specific mutation of 5mC to T. The subsequent detection was performed using the tracrRNA+DR+Septin9 Spacer strategy. As shown in Supplementary Fig. 13, successful detection (as low as 0.05%, 5mC/C) of the high methylation of the Septin9 gene was achieved, indicating that our approach not only detects different viral nucleic acid sequences but also identifies 5 mC, the potential DNA modification biomarker, by simply modifying the Spacer Region. Future efforts will focus on detecting clinical samples.

## Precisely regulating Cas12b activity through temperature or time-based manipulation of the glyoxal-caged DR region

The DR region in our split sgRNA (tracrRNA+DR+Spacer) represents the shortest universal sequence (11 nt), which is indispensable for its function (Supplementary Fig. 4a, Lane 11). Moreover, short RNAs are easier to chemically modify than longer RNAs[41]. In order to achieve precise control over the Cas12b cleavage activity, we employed glyoxal labeling of the universal DR region. Glyoxal is a well-established reagent known for its effective reaction with RNA and inhibition of RNA-enzyme interactions[42,43]. Notably, the glyoxal labeling is reversible and can be eliminated by adjusting temperature or time, thereby restoring the original functionality of the RNA[43]. We hypothesized that the glyoxal-caged DR could temporarily suppress the activity of the CRISPR-Cas12b system, which could subsequently be restored by briefly heating at 60 °C or by prolonged incubation at 37 °C, thereby recovering the system's cleavage activity, respectively (Fig. 4a).

To evaluate the effectiveness of the glyoxal-caged DR in inhibiting the CRISPR-Cas12b system, we conducted incubation experiments with DR and glyoxal for various durations (0-60 min). Gel electrophoresis analysis indicated the formation of glyoxal-DR adducts (Supplementary Fig. 14a, b). The resulting caged DR samples were then purified and assessed for Cas12b *trans*-cleavage activity. Our findings demonstrated a significant decrease in *trans*-cleavage activity after a 15-minute incubation with glyoxal, with nearly complete inhibition achieved after 45 minutes (Supplementary Fig. 14c).

Using AlphaFold3, we predicted the 3D structure of the Cas12b-split sgRNA complex, and MD simulations indicated that the glyoxal-DR adduct disrupts the assembly of split sgRNA with Cas12b, thereby inhibiting its activity (Supplementary Fig. 14d). Subsequently, we utilized the 45-minute glyoxal-caged DR to conduct recovery experiments. Gel electrophoresis indicated that the glyoxal-DR adduct rapidly dissociates upon heating, restoring the original bands (Supplementary Fig. 15a). The caged DR was subjected to heating at 60 °C for different periods (0–40 min). As depicted in the Fig. 4b, we observed a gradual recovery of Cas12b *trans*-cleavage activity after 5 minutes of heating, with full restoration of functionality achieved after 40 minutes (Fig. 4b, c). To further illustrate the recovery of activity, we employed a FAM-labeled EBV target to demonstrate the presence of *cis*-cleavage activity. Agarose gel analysis revealed a gradual recovery of *cis*-cleavage activity after 5 minutes of heating at 60 °C, reaching the same level as that of the unmodified DR after 40 minutes (Supplementary Fig. 16b). Similarly, incubating the glyoxal-caged DR at physiological temperature (37°C) showed gradual and reversible restoration of Cas12b activity. As depicted in Supplementary Fig. 16d, *trans*-cleavage activity noticeably recovered after 3 hours of incubation at 37 °C and was nearly completely restored after 24 hours. Agarose gel analysis suggested similar results for *cis*-cleavage activity, with partial recovery observed after 3 hours of incubation and near-complete recovery achieved between 12 and 24 hours, comparable to

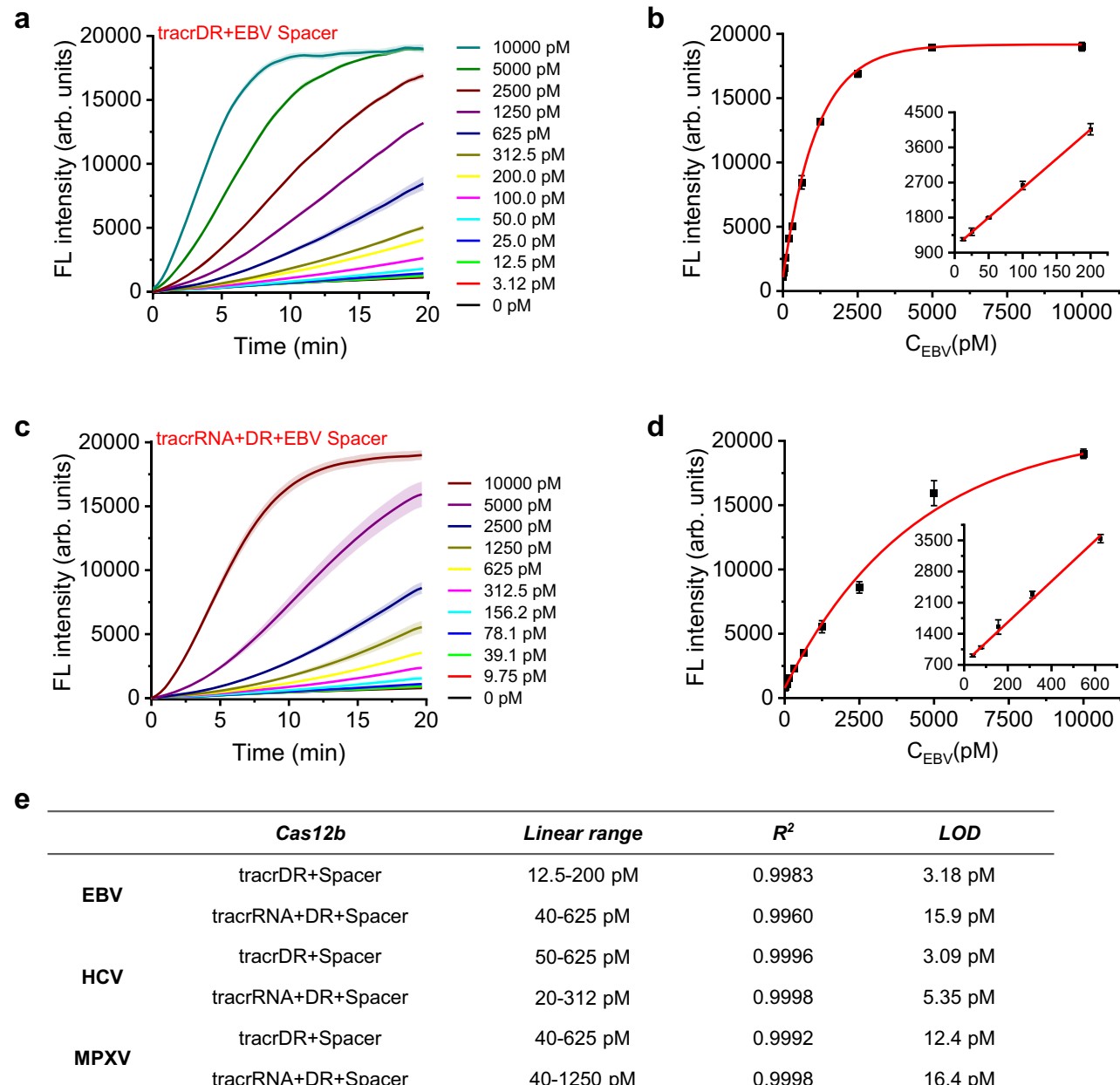

**Fig. 3 | The tracrDR+Spacer and tracrRNA+DR+Spacer strategies are employed to detect distinct nucleic acid targets through simple modifications to the Spacer region.** The tracrDR+Spacer strategy was tested with EBV-specific Spacers against various concentrations of EBV targets, yielding fluorescence detection values (a) and their corresponding linear relationship plot (b). Data are shown as mean value +/− SD ($n = 3$, biologically independent samples). Similarly, the tracrRNA+DR+Spacer strategy was assessed using the EBV Spacer with varying concentrations of EBV targets, as indicated by the fluorescence detection values (c) and the corresponding linear relationship plot (d). Data are shown as mean value +/− SD ($n = 3$, biologically independent samples). **e** A summary table detailing the linear ranges and detection limits for different targets (EBV, HCV, MPXV) as determined by the tracrDR+Spacer and tracrRNA+DR+Spacer strategies, accomplished by simply replacing the relevant Spacer regions. Source data are provided as a Source Data file.

the unmodified DR (Supplementary Fig. 16c). However, the full-length sgRNA exhibited inferior performance, experiencing significant inhibition of Cas12b *trans*-cleavage activity after a 25-minute reaction with glyoxal, with complete inhibition occurring after 45 minutes (Supplementary Fig. 15b). Moreover, even under the condition of a 40-minute heating at 60 °C, the 45-minute glyoxal-caged sgRNA demonstrated very limited ability (3.98% recovery rate) to restore Cas12b functionality (Supplementary Fig. 15c). And the 25-minute glyoxal-caged sgRNA achieved only about 60.3% recovery (Supplementary Fig. 15c). In contrast, the glyoxal modification of the shorter DR sequence enabled more precise regulation of Cas12b function (Fig. 4b, c, Supplementary Fig. 16).

### Light-controlled one-pot CRISPR-Cas12b clinical nucleic acid detection system by PC linker conjugated DR

In traditional one-pot isothermal amplification-CRISPR-Cas nucleic acid detection reactions, the Cas protein begins its *cis*-cleavage activity before the nucleic acid template has completed amplification, often leading to template degradation. This incompatibility is particularly pronounced in the detection of clinical samples, where the target concentrations are low[24–26]. A promising solution is to temporarily inactivate the Cas protein and subsequently reactivate it after amplification is complete. This approach has shown considerable success in the regulation of Cas12a[24–26,33,44–46]. However, research on the precise light-controlled regulation of Cas12b activity remains lacking,

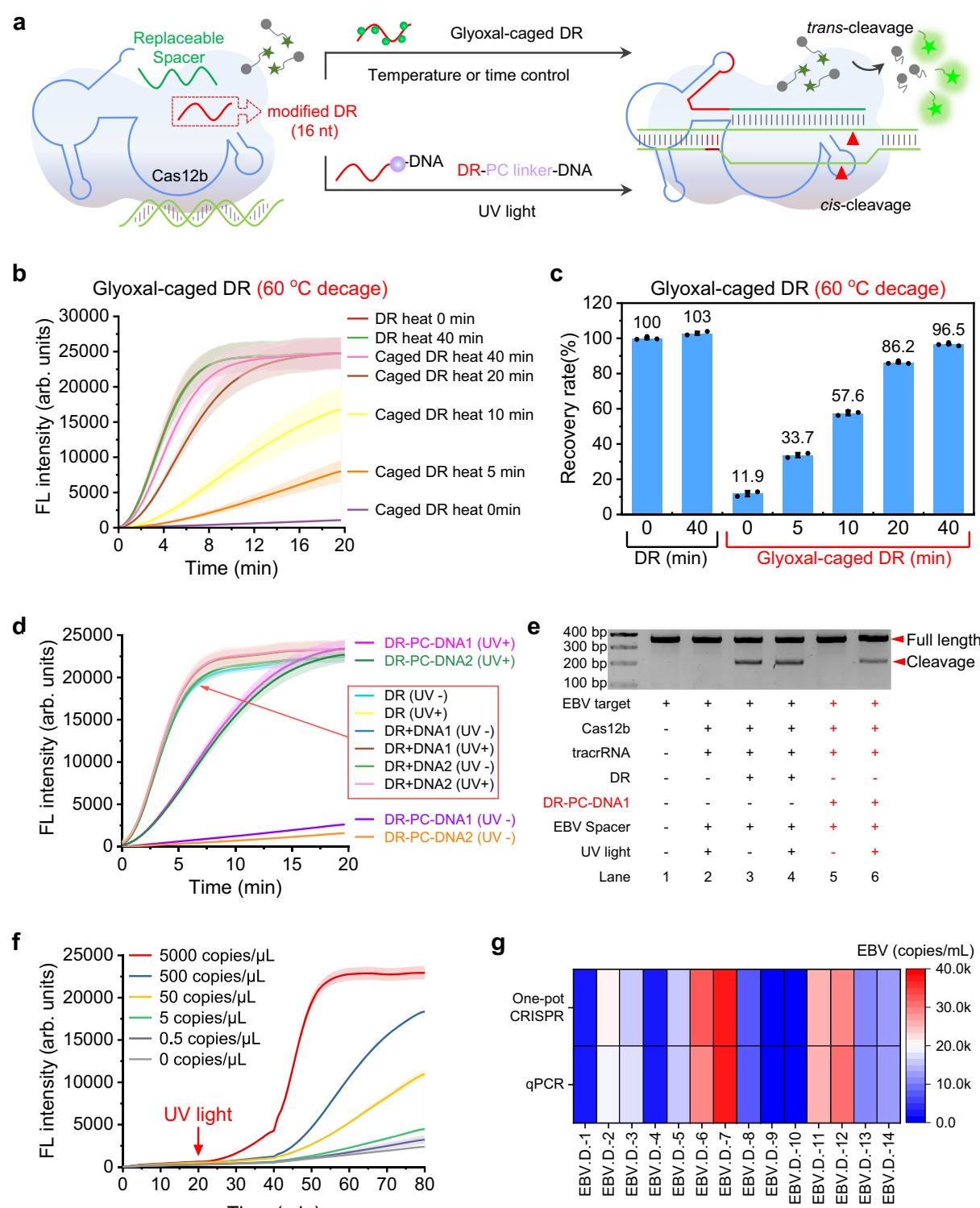

primarily due to the limitations in the length of sgRNA associated with Cas12b.

In our study, we propose that connecting a commercially available PC linker and several DNA bases to the short, universally applicable, and easily modifiable DR region of the split sgRNA (DR-PC-DNA) can effectively and temporarily inhibit Cas12b activity (Fig. 4a). Following this, UV light exposure can conveniently induce the breakage of the PC

linker, thereby restoring Cas12b activity. To evaluate the inhibitory effect of DR-PC-DNA on the CRISPR-Cas12b system, we synthesized DR-PC-DNA1 and DR-PC-DNA2 (Supplementary Table 1). UV exposure experiments (365 nm, 35 W) indicated cleavage of the PC linker within 15 seconds, with the resulting product reverting to a structure consistent with the canonical DR (Supplementary Fig. 17). Subsequently, we demonstrated the efficient suppression of Cas12b *trans*-cleavage

**Fig. 4 | Precise regulation of the CRISPR-Cas12b system by modification of the universal DR in the currently developed split sgRNA. a** Schematic representation of how glyoxal-caged DR effectively inhibits both the *cis*- and *trans*-cleavage activities of the CRISPR-Cas12b system in the tracrRNA+DR+Spacer configuration. This function can be precisely restored through temperature- or time-mediated regulation. An alternative approach illustrated here employs a PC linker attached to the DR region, which similarly inhibits the *cis*- and *trans*-cleavage activities of CRISPR-Cas12b, with restoration achieved via UV exposure. **b** Real-time fluorescence data demonstrate the extent of inhibition on *trans*-cleavage activity by glyoxal-caged DR, showing complete recovery upon heating. A 45-minute exposure to glyoxal-caged DR significantly reduces *trans*-cleavage activity (0 min). Following a 5-minute incubation at 60 °C, the activity gradually recovers, reaching near-complete restoration by 40 minutes. Data are shown as mean value +/− SD (*n* = 3, biologically independent samples). **c** A bar chart illustrates the correlation between incubation time and the recovery rate of *cis*-cleavage activity for glyoxal-caged DR under 60 °C heating conditions. All the experiments were conducted in triplicate, and error bars represent mean value +/− SD (*n* = 3). **d** Real-time fluorescence data suggest the inhibitory effect of an 8 nt random DNA sequence attached to the DR region on Cas12b's *trans*-cleavage activity, with restoration observed after 15 seconds of UV light exposure. Importantly, this 15-second UV exposure does not adversely affect the system, and the presence of 8 nt random DNA (DNA1, DNA2) does not disrupt its function. Data are shown as mean value +/− SD (*n* = 3, biologically independent samples). **e** Gel electrophoresis results provided supporting evidence for the effective inhibition of CRISPR-Cas12b's *cis*-cleavage activity by DR-PC-DNA, with activity recovery observed following 15 seconds of UV light exposure. The experiment was performed three times. **f** Real-time fluorescence data illustrating the detection of varying copy numbers of EBV using the developed one-pot RPA-CRISPR-Cas12b method. Following a 20-minute RPA reaction, Cas12b activity is activated via 15 seconds of UV irradiation (365 nm, 35 W), resulting in the release of fluorescence signals. Data are shown as mean value +/− SD (*n* = 3, biologically independent samples). **g** A heat map comparing results from traditional qPCR methods with those from the developed split sgRNA-assisted one-pot RPA-CRISPR-Cas12b methods applied to different clinical plasma samples from EBV-infected individuals. Error bars represent standard deviation (*n* = 3). Data are shown as mean ± standard deviation, based on three technical replicates. Source data are provided as a Source Data file. This Fig. 4a is adapted from Wang, J., Zhang, W., Li, W., Xie, Q., Zang, Z. and Liu, C. Enhancement of CRISPR-Cas12a system through universal circular RNA design. *Cell Rep. Methods* 5, 101076 (2025).[62], licensed under CC-BY 4.0 (https://creativecommons.org/licenses/by/4.0/).

activity by DR-PC-DNA1/2. Comparative analysis in Fig. 4d showed that the introduction of random sequence DNA1/2 alone did not affect Cas12b activity. However, when DNA1/2 was connected directly to the DR region via the PC linker, the *trans*-cleavage activity of Cas12b was almost completely inhibited. We then restored Cas12b activity by irradiating DR-PC-DNA1/2 with UV (365 nm, 35 W) for 15 seconds, successfully eliminating the inhibitory effect (Fig. 4d, Supplementary Fig. 18a). Furthermore, comparison of unmodified DR under UV irradiation suggested that 15 seconds of UV exposure did not detectably alter Cas12b functionality (Fig. 4d, Supplementary Figs. 17, 18c).

Subsequently, we employed agarose gel experiments to investigate the inhibitory effect of the DR-PC-DNA strategy on Cas12b *cis*-cleavage activity. Replacement of DR with DR-PC-DNA led to complete suppression of Cas12b *cis*-cleavage activity at 37 °C (Fig. 4e) and 48 °C (Supplementary Fig. 18c). By subjecting them to UV (365 nm, 35 W) irradiation for 15 seconds, we observed a reactivation of Cas12b *cis*-cleavage activity with a recovery rate of 56.9% at 37 °C (Supplementary Fig. 18b) and 61.3% at 48 °C (Supplementary Fig. 18d). Building upon these findings, we developed a light-controlled one-pot RPA-CRISPR-Cas12b nucleic acid detection system based on DR-PC-DNA (Supplementary Fig. 19a). The incompatibility issue between the RPA and CRISPR reaction was effectively resolved by combining the RPA-CRISPR-Cas12b system with DR-PC-DNA. DR-PC-DNA inhibits Cas12b's cleavage activity, preventing interference with the RPA. After the RPA process is complete, a 15-second UV irradiation is applied to restore Cas12b cleavage activity, enabling the collection of the fluorescent signal (Fig. 4f).

To further evaluate our approach, 14 blood samples from patients infected with EBV, as well as 12 samples from healthy donors were collected. DNA extraction from these blood samples was performed, and the RPA-CRISPR-Cas12b detection system, utilizing tracrRNA+DR-PC-DNA+Spacer as described above, was applied. The copy number of EBV in the clinical samples was determined (Fig. 4g, Supplementary Figs. 19d, 20) using the standard curve established by this system (Supplementary Fig. 19b, 19c). Comparison with the gold standard qPCR technique (Supplementary Figs. 21, 22) revealed exceptional agreement, achieving close concordance and improved detection limits (Fig. 4g, Supplementary Figs. 19d, 20, 21, 22).

## MicroRNA as Spacer region in the split sgRNA strategy for Cas12b

In our split strategy, the length of the Spacer Region closely resembles that of microRNA. This prompted us to hypothesize that microRNA could replace the Spacer Region in the Split sgRNA (either tracrDR +Spacer or tracrRNA+DR+Spacer), thereby preserving the functionality of Cas12b. This approach will allow the design of appropriate DNA activators to target specific microRNAs, facilitating direct detection of microRNAs without the need for reverse transcription or amplification (Fig. 5a).

To test our hypothesis, we focused on miR-21 and designed its corresponding DNA activator (Supplementary Table 3). The reaction was prepared with tracrRNA, DR (or tracrDR), the DNA activator for miR-21, F-Q, Cas12b, and the necessary buffers. We then introduced either miR-21 or other microRNAs such as miR-17, 31, 92a, 429, and 4429, along with the precursor miR-21. Our results indicated that only miR-21 effectively formed a complex with tracrRNA, DR (or tracrDR), and Cas12b to recognize the activator, thus activating the *trans*-cleavage activity of Cas12b to cleave F-Q and release fluorescence. In contrast, the other microRNAs and the precursor miR-21 did not elicit this response. This demonstrates that our split method using tracrDR or tracrRNA+DR effectively detects miR-21, while other tested micro-RNAs and the precursor (Supplementary Fig. 23a) fail to do so (Fig. 5b, Supplementary Fig. 23b, c, e, f).

Subsequently, we replaced F-Q with FAM-Biotin (Supplementary Table 1) and applied the method to commercial CRISPR paper test strips. Our results indicated that only miR-21 led to the appearance of the T line, while the other tested microRNAs, the precursor miR-21, and the control remained constant in the C line. This further provided supporting evidence for the high selectivity of our microRNA detection method. (Supplementary Fig. 23d, g). We then designed and synthesized corresponding activators for miR-21, miR-17, miR-31, and miR-92a, all of which are potential biomarkers for colorectal cancer. By pairing tracrDR with these activators, we tested a range of concentrations of microRNA standards. Our findings revealed that the fluorescence signal from the system increased in proportion to the concentrations of these microRNAs, exhibiting a linear relationship within a certain range. The limits of detection for miR-21, miR-17, miR-31, and miR-92a were determined to be 26.7 fM, 37.2 fM, 33.5 fM, and 28.9 fM, respectively (Supplementary Fig. 24). Moreover, the use of commercial CRISPR test strips enabled the reliable detection of various microRNA concentrations as low as 50 fM (Fig. 5c, d, Supplementary Fig. 25). This novel approach, which employs microRNA as the Spacer Region in our currently developed Split sgRNA Strategy, shows considerable promise for the direct detection of microRNAs without the need for reverse transcription or amplification.

To further explore the applicability of our split strategy, we investigated its responsiveness to single-nucleotide mutations. Previous studies have reported that Cas12b can respond to such mutations. We hypothesized that our split strategy would enhance Cas12b's sensitivity to single-nucleotide variants[3,47]. As demonstrated in the data presented in Fig. 5e, single-nucleotide mutations (A-T, T-A, G-C, or C-G)

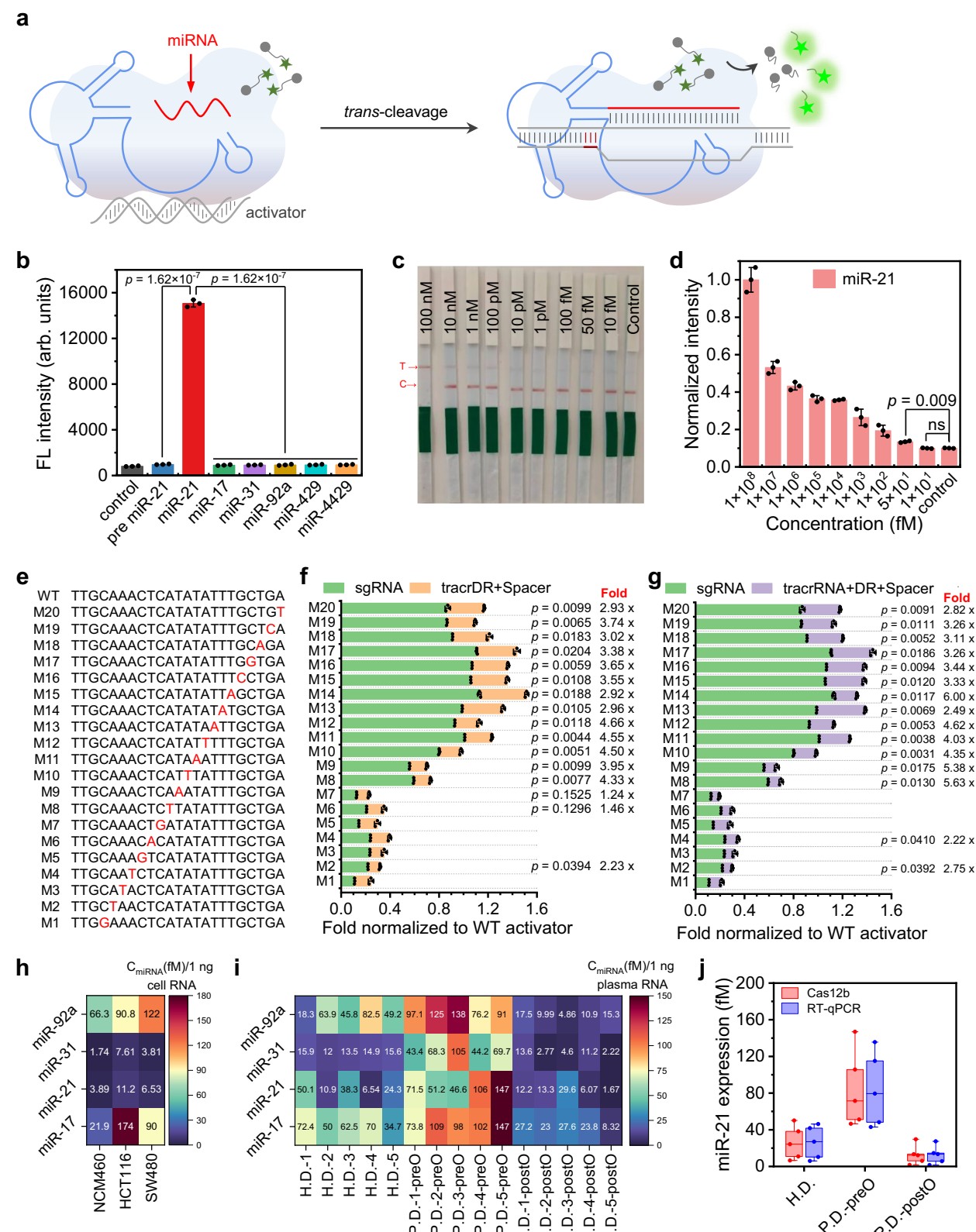

were introduced at various positions of the activator, specifically at the nearest PAM 1-20 position (M1-20, Supplementary Table 4), while preserving the GC content of the double-strand. The results obtained indicated that at the nearest PAM 1-7 position (M1-7), both the full-length sgRNA and the split sgRNA exhibited lower fluorescence signals in comparison to the unmutated wild-type (Fig. 5f, g, Supplementary Fig. 26). This suggests that mutations at M1-7 are effectively detected

by all sgRNA forms. Notably, at position M2, the fluorescence signal from the full-length sgRNA was 2.23 times (Fig. 5f) that of the tracrDR +Spacer approach and 2.75 times (Fig. 5g) that of the tracrRNA+DR +Spacer, indicating heightened sensitivity of our split strategy to single-nucleotide mutations. Conversely, the original tracrRNA+crRNA strategy demonstrated diminished discrimination (0.64 times) (Supplementary Fig. 26a) compared to the full-length sgRNA. At positions

**Fig. 5 | Detection of microRNA using the mechanism of microRNA as the spacer region in the split sgRNA strategy for Cas12b. a** Schematic representation illustrating how microRNA can replace the original Spacer of split sgRNA, thereby guiding the activity of Cas12b. **b** Experiments supporting the selective detection of different microRNAs using the split strategy paired with a DNA activator specific to miR-21. Notably, other microRNAs, as well as the precursor of miR-21, do not yield fluorescence. Error bars indicate the mean value +/− SD (*n* = 3) of biologically independent samples and statistical analysis was conducted using a two-tailed t-test. **c** Detection of microRNA using commercial paper test strips with various concentrations of miR-21. The photograph shows the test strip results, where the T line indicates the presence of miR-21 and the C line serves as a control. All the experiments were conducted in triplicate and error bars represent mean value +/− SD (*n* = 3). **d** Schematic representation of ImageJ quantitative analysis applied to the results shown in (**c**). Error bars indicate the mean value +/− SD (*n* = 3) of biologically independent samples, and statistical analysis was conducted using a two-tailed t-test. **e** Detailed design for evaluating Cas12b response to single-nucleotide mutations using the split strategy. WT refers to the wild-type DNA activator sequence, which is devoid of mutations. The first three bases, TTG, are Cas12b PAM sequence, and the rest 20 bases following the PAM correspond to the Spacer region of Cas12b. The red-highlighted bases indicate mutation sites, with M20 denoting a mutation at the 20th base from the 3′ end of PAM, and M1 corresponding to a mutation at the 1st base from the same end. Single-nucleotide mutations (A-T, T-A, G-C, or C-G) were designed to maintain a constant GC content within the double-stranded DNA activator. **f** Comparison of fluorescence readings for full-length sgRNA versus split sgRNA (tracrDR+Spacer) when detecting wild-type and single base mutations (M1-20). The y-axis represents the mutation position, correlating with (**e**), while the x-axis indicates the ratio of fluorescence values obtained from the mutated sequences compared to the wild type. Green bars represent full-length sgRNA,

while brown bars represent the tracrDR+Spacer configuration. Asterisks on the right (*,**) signify significant differences between full-length sgRNA and tracrDR +Spacer, with the accompanying numbers indicating the fold change between them. Error bars indicate the mean value +/− SD (*n* = 3) of biologically independent samples and statistical analysis was conducted using a two-tailed t-test. **g** Similar to (**f**), this figure compares fluorescence values of full-length sgRNA and split sgRNA (tracrRNA+crRNA+Spacer) in detecting wild-type and M1-20 single base mutations. Error bars indicate the mean value +/− SD (*n* = 3) of biologically independent samples, and statistical analysis was conducted using a two-tailed t-test. **h** The tracrDR matches with DNA activators corresponding to miR-17, 21, 31, and 92a, assessing the levels of these microRNAs in total RNA extracted from colorectal cancer cell lines HCT116 and SW480, as well as non-cancerous intestinal cell line NCM460, within the developed CRISPR-Cas12b system. Different colors represent varying concentrations, with specific values provided in the boxes. **i** Results of miR-17, 21, 31, and 92a concentration detection in total RNA from the plasma of healthy donors (H.D.-1-5), colorectal cancer patients pre-operation (P.D.-1-5 preO), and post-operation (P.D.-1-5 postO). Different colors denote distinct concentrations, with specific values shown in the boxes. **j** Comparison of the miR-21 concentration detected in total RNA from the plasma of healthy donors (H.D.-1-5), preoperative colon cancer patients (P.D.-1-5 preO), and postoperative patients (P.D.-1-5 postO) using our current Cas12b split sgRNA approach (tracrDR+miR-21 corresponding DNA activator) versus traditional RT-qPCR methods. The median expression level is depicted by the center line, the interquartile range by the box boundaries, and the whisker extrema represent the maximum and minimum values. Source data are provided as a Source Data file. This Fig. 5a is adapted from Wang, J., Zhang, W., Li, W., Xie, Q., Zang, Z. and Liu, C. Enhancement of CRISPR-Cas12a system through universal circular RNA design. *Cell Rep. Methods* 5, 101076 (2025)[62], licensed under CC-BY 4.0 (https://creativecommons.org/licenses/by/4.0/).

M8-20, the capacity of the full-length sgRNA to discern single-nucleotide mutations was suboptimal, while our split strategy exhibited a marked improvement (approximately 3-5 times greater fluorescent divergence) over both the full-length sgRNA and traditional tracrRNA+crRNA methods (Fig. 5f, g, Supplementary Fig. 26). These findings provide evidence that our split strategy enhances Cas12b's detection capabilities for single-nucleotide mutations.

Following this, we cultured colorectal cancer cell lines HCT116 and SW480, as well as a non-cancerous intestinal cell line NCM460. We extracted microRNA from these cultured cells and utilized our method to detect miR-17, miR-21, miR-31, and miR-92a, with the results further supported by conventional RT-qPCR. We observed that the expression levels of microRNAs in the non-cancerous NCM460 were significantly lower than those in cancerous cell lines HCT116 and SW480 (Fig. 5h, Supplementary Fig. 27). The quantitative results from our method closely aligned with those obtained from conventional RT-qPCR, demonstrating the strong potential and performance of our split strategy for microRNA detection (Fig. 5h, Supplementary Figs. 27–30).

We also collected plasma samples from five healthy donors and five colorectal cancer patients before and after surgery. Using both our developed method and conventional RT-qPCR, we detected miR-17, miR-21, miR-31, and miR-92a in the plasma samples. Our results indicated that the microRNA levels in healthy individuals were lower than those of colorectal cancer patients pre-operation but slightly higher than those of patients post-operation (Fig. 5i). Results from our detection method closely corresponded with those obtained from conventional RT-qPCR (Fig. 5j, Supplementary Fig. 31). Our data provided supporting evidence for the potential of miR-17, miR-21, miR-31, and miR-92a as biomarkers for diagnosing colorectal cancer and monitoring post-operative recovery, while also demonstrating the utility of our method in microRNA detection. The strategy of utilizing microRNA as a Spacer in the split sgRNA approach for CRISPR-Cas12b is both feasible and holds great promise for future applications.

## Discussion

The CRISPR-Cas12b system has garnered increasing attention for its inherent advantages in molecular diagnostics, underlining its potential

significance in clinical settings. Compared to its counterpart Cas12a, Cas12b exhibits several superior properties, including enhanced thermostability[7,9], broader pH tolerance[7], and greater stability under physiological conditions[7]. These properties make it particularly suitable for high-temperature applications such as LAMP[13] and gene editing in tropical crops[11]. And Cas12b's strong PAM preference (e.g., TTN) can narrow its targeting range compared to certain Cas12a variants, potentially reducing off-target effects[5]. In addition, Cas12b exhibits superior single-base resolution and target recognition specificity, further enhancing its precision[3].

Despite these advantages, Cas12b also has notable limitations. For example, Cas12a requires only about 42 nt of crRNA for functionality, and significant advances have been made in crRNA modifications for Cas12a. These include elongation[48,49], truncation[21], splitting[30–32,50], introduction of secondary structures[51], DNA embedding[21,49], and substitution of natural[52–54] or synthetic[24,55,56] modified nucleotides, as well as incorporation of complementary sequences[18,25,44,57–59]. These innovations have provided effective means to precisely regulate Cas12a activity. In contrast, Cas12b requires sgRNAs longer than 100 nt, which complicates the chemical modification of RNA for activity regulation and limits its broader application. Consequently, the chemical modification of Cas12b sgRNA and precise control of its activity remain largely unexplored, significantly limiting its utility. This gap highlights the importance of our study on split sgRNA strategies for Cas12b.

Our findings demonstrate the feasibility of splitting Cas12b sgRNA, exceeding 100 nucleotides in length, into two or three segments. Our experimental results support the conclusion that split sgRNA retains the ability to guide Cas12b in both *cis*- and *trans*-cleavage activities. We explored the reduced affinity of split sgRNAs for Cas12b compared to full-length sgRNAs, as evidenced by an increased dissociation constant. While the *cis*-cleavage activity of split sgRNAs is relatively diminished, the larger dissociation constant of the split sgRNA-Cas12b complex with target DNA facilitates easier dissociation of the target. Since *cis*- and *trans*-cleavage share the same catalytic site, the *trans*-cleavage activity is consequently enhanced. This hypothesis aligns with our observation that our split sgRNAs yield fewer *cis*-cleavage products but maintain comparable *trans*-cleavage activity. The

dissociation constant of the sgRNA-Cas protein complex for target DNA may directly correlate with *trans*-cleavage activity, with a larger dissociation constant resulting in relatively stronger *trans*-cleavage activity.

Importantly, this split sgRNA approach allows flexible targeting of different nucleic acids by simply replacing the Spacer region. By circumventing the limitations associated with traditional sgRNA design, our method offers a streamlined solution: changing only the Spacer (18-22 nt) can facilitate the detection of different targets. Incorporating the universal component into the split strategy reduces design complexity and associated diagnostic costs.

Furthermore, we explored the potential of glyoxal-labeled universal DR in split sgRNA to modulate Cas12b activity. Our results reveal that glyoxal caged DR effectively inhibits Cas12b, while de-labeling through heating or extended incubation restores its activity. This finding provides a mechanistic basis for the dynamic regulation of CRISPR activity, enabling programmable responses to specific stimuli. Additionally, we connected a commercial PC linker and a random DNA sequence to the DR, successfully inactivating Cas12b. By applying UV light to cleave the PC linker, we were able to reactivate Cas12b functionality. Our approach seamlessly integrates isothermal amplification strategies, particularly RPA, with CRISPR reactions, achieving successful one-pot detection of trace EBV in human plasma samples with sensitivity and specificity comparable to traditional qPCR techniques. This represents a significant advancement in the simplicity and speed of diagnostics, critical for real-time clinical applications.

Moreover, we uncover that microRNA can serve as a substitute for the Spacer region in guiding Cas12b cleavage, corroborating findings by Y. Chen et al. in Cas12a split crRNA[30]. This breakthrough in microRNA detection is noteworthy; we successfully identified microRNA in cultured cell samples, as well as in plasma from healthy individuals and pre-operative and post-operative colorectal cancer patients. Our results align with existing RT-qPCR methods, while reducing procedural steps and eliminating the need for reverse transcription and amplification, establishing our method as a compelling alternative for microRNA analysis. Furthermore, the analysis of microRNA panels in both colorectal cancer cell lines and patients shows considerable promise for prognostic monitoring post-surgery. Notably, in our microRNA detection designs, we used double-stranded DNA activators (target DNA) with an optimal PAM sequence, while single-stranded DNA activators showed no significant difference in LOD. Although Cas proteins traditionally recognize single-stranded nucleic acids more effectively, when DNA activators are in large excess, their single- or double-stranded nature becomes less critical for LOD. Using split sgRNA and Cas12b alone, we achieved an LOD of ~10,000 fM for target DNA. However, by using split sgRNA (without the Spacer region) and target DNA to detect microRNAs, the LOD dropped to 100 fM - a 100-fold improvement. This improvement may be attributed to the significantly lower dissociation constant of Cas12b for sgRNA (split or full-length) compared to that of the Cas12b-sgRNA complex for target DNA, indicating a stronger affinity of Cas12b for sgRNA than the Cas12b-sgRNA complex for target DNA. Despite the existence of numerous excellent microRNA CRISPR detection methods[30,60,61], our mechanistic exploration underscores the significant value of our split sgRNA strategy for Cas12b. Our results enhance the capability of Cas12b for direct microRNA detection, providing a promising advancement in the field.

And our approaches also have limitations. The split sgRNA exhibits reduced affinity for Cas12b, which enhances *trans*-cleavage activity but compromises *cis*-cleavage efficiency, limiting its use in applications requiring robust *cis*-cleavage. While the split strategy simplifies sgRNA design and allows direct detection of the microRNA as Spacer without reverse transcription or amplification, its sensitivity to target DNA remains suboptimal, particularly for trace clinical samples, necessitating integration with isothermal amplification methods. Although Cas12b demonstrates strong discrimination of single base

mutations[3] and our split strategy further improves this capability, detection efficiency varies depending on the position of the mutation within the Spacer, requiring additional optimization for clinical applications. Finally, small molecule modifications to the universal DR for precise regulation of Cas12b activity increase experimental complexity and cost, a challenge that could be mitigated by future commercialization of modified RNA components.

In summary, Cas12b demonstrates several superior properties, including enhanced thermostability, broader pH tolerance, and greater stability under physiological conditions, making it particularly well-suited for high-temperature applications. Additionally, Cas12b's strong PAM preference (e.g., TTN) narrows its targeting range compared to certain Cas12a variants, which may reduce off-target effects. Furthermore, Cas12b exhibits superior single-base resolution and target recognition specificity, further enhancing its precision. However, Cas12b also has several limitations compared to Cas12a. Notably, Cas12a functions efficiently with ~42 nt crRNA and has been optimized for crRNA modifications. In contrast, Cas12b requires significantly longer sgRNA (>100 nt). The increased length complicates the chemical modification of sgRNA, which hinders precise control of Cas12b's cleavage activity and restricts its broader application. Here, we highlight the efficacy of the split sgRNA strategy in precisely regulating Cas12b functionality and expanding its application scope. Utilizing one to two universal components along with replaceable Spacer presents a modular design, promising to enhance nucleic acid detection across various clinical and research contexts. Future investigations will focus on further optimizing this platform for clinical evaluation and exploring its integration into broader diagnostic workflows.

## Methods
### Ethical statement
Human blood samples were collected and provided by the Seventh Affiliated Hospital, Sun Yat-sen University, with protocols approved by the ethics committee at the Seventh Affiliated Hospital, Sun Yat-sen University (KY-2024-192-01, KY-2024-384-02, KY-2024-009-02, and KY-2023-117-01). All patients in this study signed an informed consent form. All human clinical samples were collected in a randomized manner without further stratification based on gender. The clinical samples in this proof-of-concept study were collected randomly without intentional stratification by sex or gender. As this work primarily focuses on establishing technical feasibility for detecting trace nucleic acid biomarkers using our novel split sgRNA design, the preliminary analysis did not include sex/gender comparisons.

### Chemicals and materials
All chemicals were purchased from Bide Pharmatech Co., Ltd. (Shanghai, China), except where otherwise specified. The Cas12b protein (AapCas12b, Cat No. CAS-12P-100) was expressed and purified from E. coli by EZassay Biotech (Shenzhen, China). RNA and DNA were synthesized and purified by IGE Biotech (Guangzhou, China), Accurate Biotechnology (Hunan) Co., Ltd. (Changsha, China), or Generay Biotechnology (Shanghai, China). RT-RPA 2.0 (Cat No. 901221021), Lateral flow detection paper strips (Cat No. CAS-cmCSA01) were supplied from Keer Life (Suzhou, China). Fetal bovine serum (Cat No. 209111) was procured from NEST Biotechnology Co., Ltd. (Wuxi, China). GelstainRed™ Nucleic Acid Dye (Cat No. S2009L) was purchased from US EVERBRIGHT (Suzhou, China). Hieff Canace® Plus High-Fidelity DNA Polymerase (Cat No. 10153ES76), T4 polynucleotide Kinase (Cat No. 12902ES76), Proteinase K solution (Cat No. 10412ES76), MolPure® PCR Purification Kit (Cat No. 19106ES70), MolPure® Cell/Tissue Total RNA Kit (19221ES50), MolPure® Blood RNA Kit (19241ES50) were bought from Yeasen Biotechnology Co., Ltd. (Shanghai, China). QIAamp DNA Blood Midi Kit (Cat No. 51183) was bought from Qiagen (USA). miRNA 1st Strand cDNA Synthesis Kit (by stem-loop) (MR101-01) was purchased from Vazyme (Nanjing, China).

## Assessment of CRISPR-Cas12b *Trans*-cleavage activity via denaturing polyacrylamide gel electrophoresis (PAGE) and fluorescent signal detection

We prepared a 10X Cas12b reaction buffer containing 20 mM spermidine, 400 mM Tris-HCl, 400 mM glycine, 60 mM MgCl$_2$, 10 mM DTT, 0.01% Triton X-100, 4% PEG-200, pH 8.5. The buffer was aliquoted and stored at −20 °C for future use, ensuring that only new, single-use aliquots were thawed to minimize contamination and interference. In a 200 µL PCR tube, we added 1 µL of the 10X Cas12b reaction buffer, 1 µL of EBV target (200 nM, either single-stranded or double-stranded), 1 µL of FAM-ssDNA (2500 nM, as a single-stranded FAM-labeled DNA cleavage reporter), 1 µL of Cas12b (2000 nM), and varying concentrations of either full-length or split sgRNA (including EBV sgRNA, tracrRNA+EBV crRNA, tracrDR+EBV Spacer, or tracrRNA+DR + EBV Spacer). The total volume was brought to 10 µL with ddH$_2$O and mixed thoroughly. The reaction mixture was incubated at 48 °C for 15 minutes, followed by the addition of 20 µL of formamide (deionized) to terminate the reaction. The products were then analyzed by denaturing PAGE (20% acrylamide gel containing 7 M urea) at a voltage of 380 V for 40 minutes. Gel imaging was performed to visualize the results. In the fluorescent signal detection, we substituted the FAM-ssDNA with F-Q while keeping the other experimental conditions unchanged. The reaction mixture was incubated at 48 °C for 20 minutes in a qPCR instrument (YEASEN, Celemetor Real-Time Fluorescent Quantitative PCR Analysis System-Celemetor 96), with fluorescence readings taken every 20 seconds.

## Determination of Michaelis-Menten kinetics for full-length and split sgRNA-assisted CRISPR-Cas12b system

To ascertain the kinetic parameters of the F-Q reporter, we performed serial twofold dilutions of the F-Q reporter and incubated them with 100 nM Cas12b, 100 nM sgRNA, 10 nM activators, and 1X reaction buffer at 48 °C for 30 minutes. The final fluorescent intensity was measured to establish standard curves that correlate fluorescent intensity to the concentration of cleaved and uncleaved F-Q reporter. The reactions comprised various concentrations of F-Q reporter (0, 15.6, 31.2, 62.5, 125, 250, 500, and 1000 nM), alongside 25 nM Cas12b, 100 nM of four sgRNA types (sgRNA, tracrRNA+crRNA, tracrDR +Spacer, and tracrRNA+DR+Spacer), and 1 nM activators. Continuous fluorescence detection was performed for 150 seconds at 48 °C. The reaction velocity, determined through linear fitting, was plotted against the F-Q reporter concentration to determine the Michaelis-Menten constants using Origin software.

## Assessment of CRISPR-Cas12b *Cis*-cleavage activity

In separate 200 µL PCR tubes, we added 1 µL of the 10X Cas12b reaction buffer, 1 µL of FAM-labeled EBV target (200 nM, double-stranded), 1 µL of Cas12b (2000 nM), and differing concentrations of full-length or split sgRNA (including EBV sgRNA, tracrRNA+EBV crRNA, tracrDR+EBV Spacer, or tracrRNA+DR + EBV Spacer). The mixture was brought to a final volume of 10 µL with ddH$_2$O and incubated at 48 °C for various durations. Following incubation, 40 µg of proteinase K was introduced, and the reaction was maintained at 37 °C for an additional 15 minutes to terminate cleavage activity. The CRISPR-Cas12b *cis*-cleavage activity was evaluated using 2% agarose gel electrophoresis (120 V for approximately 30 minutes).

## Microscale thermophoresis (MST) assays for dissociation constant (Kd) determination

For Kd determination of Cas12b-sgRNA interactions, 50 nM of 3′-Cy5-labeled sgRNA (sgRNA-3′-Cy5) was prepared in assay buffer containing 10 mM Tris-HCl (pH 7.9), 50 mM NaCl, 10 mM MgCl$_2$, and 0.5% Triton X-100 (total volume 200 µl). Cas12b protein was serially diluted from an initial concentration of 10.0 µM in the same buffer. Equal volumes (10 µl) of diluted Cas12b (ligand) and 50 nM sgRNA-3′-Cy5 (target) were mixed and loaded into Monolith standard-treated capillaries. Similar

protocols were applied for assessing interactions between Cas12b and other RNA complexes (tracrRNA+crRNA-3′-Cy5, tracrDR+Spacer-3′-Cy5, or tracrRNA+DR+Spacer-3′-Cy5).

For RNP-dsDNA target interactions, 50 nM sgRNA-3′-Cy5 and 50 nM Cas12b were pre-incubated in the aforementioned buffer. dsDNA target was serially diluted from 200.0 µM in the same buffer, and 10 µl aliquots were mixed with equal volumes of pre-formed Cas12b-sgRNA complexes. The same protocol was followed for other RNA complex variants.

Measurements were performed using Monolith Pico (Nano-Temper) with the following parameters: red laser excitation for 20 seconds, 40% MST power, and 20 °C. Data analysis was conducted using MO Affinity Analysis v3.0.5 software, with Kd values determined by fitting to the following equation (triplicate measurements):

$$F_{norm} = F_{min} + (F_{max} - F_{min}) * [L] / (Kd + [L]) \qquad (1)$$

where $F_{max}$ and $F_{min}$ represent maximum and minimum signal intensities, *[L]* denotes ligand concentration, and Kd is the dissociation constant.

## Molecular dynamics simulations

The three-dimensional structures of Cas12b in complex with four distinct split sgRNA variants (sgRNA, tracrRNA+crRNA, tracrDR +Spacer, tracrRNA+DR+Spacer) were initially predicted using AlphaFold3 (https://alphafoldserver.com/). The most optimal model was selected based on prediction quality metrics and subsequently converted to PDB format using PyMOL. Molecular dynamics simulations were performed using GROMACS 2024.5 with the amber14sb_OL15 force field. A cubic simulation box with a 2.0 nm margin was defined to ensure adequate spacing between protein atoms and box boundaries, thereby minimizing periodic boundary artifacts. Following a 100 ns MD run, trajectory analysis was conducted using GROMACS tools. Specifically, gmx rms was employed for root mean square deviation (RMSD) calculations, gmx rmsf for root mean square fluctuation (RMSF) analysis, and gmx trjconv for extracting complex conformations. Data visualization was performed using PyMOL 3.0.4 and Python 3.12.2.

## Fluorescent detection of various nucleic acid targets using a universal split sgRNA strategy with matching spacers

In a 200 µL PCR tube, we combined 2 µL of 10X Cas12b reaction buffer, 2 µL of different concentrations of nucleic acid targets (EBV, HCV, or MPXV), 2 µL of F-Q (2500 nM), 2 µL of Cas12b (2000 nM), and varying concentrations of either full-length or split sgRNA. The total volume was adjusted to 20 µL with ddH$_2$O and mixed thoroughly. The reaction mixture was incubated at 48 °C for 20 minutes in a qPCR instrument, and fluorescent readings were recorded every 20 seconds to generate a time-dependent fluorescence curve. Fluorescent values proportional to the slope of the curve were selected for further analysis, including bar chart generation, curve fitting, statistical significance testing, and limit of detection calculations.

## Detection of methylation sites using the split sgRNA-assisted CRISPR-Cas12b system

To investigate hypermethylation in the SEPT9 gene, a known biomarker for colorectal cancer, we employed the universal Split sgRNA-assisted CRISPR-Cas12b system. Different ratios of high-methylated (SEPT9-5mC-X-Target, where X represents the number of methylated CpG sites ranging from 1 to 5) and non-methylated (SEPT9-C-Target) SEPT9 gene targets were subjected to TAPS treatment, which selectively converts 5-methylcytosine (5mC) to dehydroxyuridine (DHU) while leaving cytosine (C) intact[40]. The treated targets were subsequently amplified via recombinase polymerase amplification (RPA) and utilized as templates in the Split sgRNA-assisted CRISPR-Cas12b

detection system. The experimental conditions were consistent with those outlined in section 4.

## Synthesis of Caged RNA using glyoxal

To synthesize caged RNA, we combined specific RNA components, including EBV sgRNA, tracrRNA, tracrDR, or DR (at a concentration of 2 μM), with 7.25 μL of 40% glyoxal, 25 μL of DMSO, and ddH$_2$O to achieve a final volume of 50 μL. The reaction mixture was incubated at 50 °C for various durations, after which the reaction was quenched by adding 450 μL of ice-cold ethanol. The resulting mixture was purified via traditional ice-cold ethanol precipitation methods[43]. Following reconstitution in ddH$_2$O, the concentration of caged RNA was quantified using a NanoDrop spectrophotometer and stored at −20 °C for future experiments.

## Controlled cleavage activity of CRISPR-Cas12b system using glyoxal-caged RNA

We pre-incubated various concentrations and types of caged RNA at either 60 °C or 37 °C for different time intervals. Subsequently, the caged RNA was mixed with EBV target (20 nM), Cas12b (200 nM), and F-Q (250 nM) in a 1X Cas12b reaction buffer, bringing the total volume to 20 μL. The reaction mixture was then incubated at 48 °C for 20 minutes in a qPCR instrument, with fluorescence readings recorded every 20 seconds to assess *trans*-cleavage activity. Alternatively, a FAM-labeled EBV target was used in place of the unlabeled target, eliminating the need for F-Q. The *cis*-cleavage activity was evaluated by performing 2% agarose gel electrophoresis (120 V for 30 minutes), as described in "Discussion".

## Photo-controlled one-pot RPA-CRISPR-Cas12b system for clinical EBV sample analysis using DR-PC-DNA

DNA was extracted from clinical blood samples, both infected and uninfected with EBV, using the QIAamp DNA Blood Midi Kit (Qiagen, Cat No. 51183) in accordance with the manufacturer's guidelines. Clinical DNA samples or varying concentrations of EBV target standards were then introduced into a reaction mixture. This mixture comprised primers (EBV RPA-F and EBV RPA-R, each at 250 nM, Supplementary Table 1), tracrRNA (400 nM), DR-PC-dAT (400 nM), EBV Spacer (500 nM), F-Q (250 nM), Cas12b (200 nM), ADP (1 mM), T4 polynucleotide kinase (10 U), and 9 μL of RPA mix (KEER LIFE, RPA freeze-dried microspheres were dissolved in 25 ul of ddH$_2$O and 9 ul was used for each assay) in a final 1X RPA-Cas12b buffer (composed of 0.01% Tween 20, 40 mM Glycine, 10 mM Tris-HCl, 1 mM DTT, 40 mM KCl, 32.4 mM MgCl$_2$, pH 8.5), totaling a volume of 20 μL. The reaction mixture was incubated at 37 °C in a qPCR instrument for 20 minutes, with fluorescence measurements taken every 5 minutes. Subsequent to this incubation, all samples were exposed to UV light (365 nm, 35 W) for 15 seconds, followed by an additional incubation at 37 °C for 20 minutes. Finally, the samples were transferred to an incubation temperature of 48 °C, with fluorescence readings recorded every minute. The EBV content in clinical samples was determined using standard curves generated from the fluorescent data and further analyzed using column charts and curve-fitting methodologies.

## Real-time quantitative PCR (qPCR) for clinical EBV sample detection

To investigate the reliability of the photo-controlled one-pot RPA-CRISPR-Cas12b system, traditional qPCR was performed utilizing the Hieff® 1X qPCR SYBR Green Master Mix (No Rox) (YEASEN, Cat No. 11201ES08). Each reaction consisted of 0.2 μM of each primer (EBV PCR-F and EBV PCR-R, as specified in Supplementary Table 1) and 2 μL of template DNA containing varying copy numbers (ranging from $10^2$ to $10^7$ copies) of the EBV target to establish a standard curve, or clinical samples. The total reaction volume was adjusted to 20 μL using nuclease-free water. The mixture was subjected to PCR

analysis in a qPCR instrument, starting with an initial denaturation phase at 95 °C for 3 minutes, followed by 40 amplification cycles at 95 °C for 10 seconds and 55 °C for 30 seconds. The EBV content in clinical samples was quantified based on the obtained Ct values and standard curve.

## Design of a reverse transcription-free, amplification-free CRISPR-Cas12b system for microRNA detection utilizing micro-RNA as Spacer in a split sgRNA strategy for Cas12b

In a 200 μL PCR tube, 0.2 μL of Cas12b (10 μM), 0.5 μL of F-Q reporter (10 μM), 0.2 μL of tracrRNA (10 μM), and either 0.2 μL of DR (10 μM) or 0.2 μL of tracrDR (10 μM) were combined. Additionally, 2 μL of various concentrations of microRNA activator (5 μM), 2 μL of 10× NEB 2.0 buffer, and 2 μL of the target microRNA or its precursor were added, followed by DNase/RNase-free water to adjust the total volume to 20 μL. The mixture was incubated at 48 °C for 60 minutes, with fluorescence signals collected every 30 seconds. In the test strip experiment, the original 0.5 μL F-Q reporter (10 μM) was replaced with 0.5 μL of FAM-Biotin reporter (20 μM), while all other components remained unchanged. After the 48 °C incubation for 30 minutes, the test strip was inserted into the reaction solution, after which images of the experimental results were captured using a smartphone. The intensity of the bands on the test strip was quantified using ImageJ software.

## Detection of microRNA in cultured cells and human plasma samples using a CRISPR-Cas12b system with a split sgRNA strategy

Total RNA was extracted from the NCM460, HCT116, and SW480 cell lines, as well as from human plasma samples, utilizing either the Mol-Pure® Cell/Tissue Total RNA Kit or the MolPure® Blood RNA Kit. For each reaction, 100 ng of RNA was added to a solution containing 200 nM Cas12b, 400 nM tracrDR, 1× NEB 2.0 buffer, and nuclease-free water to reach a final volume of 10 μL. The mixture was subsequently incubated at room temperature for 10 minutes. After incubation, 250 nM F-Q reporter and 500 nM activator were introduced, and the total volume was increased to 20 μL with additional nuclease-free water. Fluorescence signals were recorded at 30-second intervals at a temperature of 48 °C using the Celemetor-96 real-time PCR system. A standard curve was established correlating standard microRNA concentrations with fluorescence intensity. Finally, the microRNA concentration in the cell or plasma samples was quantified by referencing this standard curve.

## Traditional RT-qPCR method for microRNA detection

One microgram of extracted RNA was reverse transcribed into complementary DNA (cDNA) using the miRNA 1st Strand cDNA Synthesis Kit according to the manufacturer's protocol. Briefly, genomic DNA was removed by incubation at 42 °C, followed by the synthesis of cDNA at 50 °C for 15 minutes, which was then inactivated at 85 °C for 5 minutes. Subsequently, a 1:10 (v/v) dilution of the cDNA was prepared, along with specific primers corresponding to the target microRNAs at a concentration of 200 nM, and mixed with SYBR Green (Vazyme). The qPCR reaction proceeded with an initial denaturation at 95 °C for 30 seconds, followed by 40 cycles of denaturation at 95 °C for 30 seconds and annealing/extension at 60 °C for 30 seconds. A standard curve was generated to correlate microRNA concentration with Cq values, allowing for the quantification of specific microRNAs in the samples based on this standard curve.

## Data processing and analysis

Data are shown as mean value +/− SD ($n = 3$) of biologically independent samples. Data analysis and processing were performed using GraphPad Prism 9.0. Statistical significance was assessed using a two-tailed t test to compare differences across various conditions. Fluorescence intensity was calculated as the difference between the final and initial fluorescence values, expressed in arbitrary units (arb.

units). For the determination of the linear detection range, a series of standard target concentrations was prepared using a 2-fold serial dilution. The linear range was identified by plotting the fluorescence signal values (y-axis) against the corresponding target concentrations (x-axis) and performing linear regression analysis. The concentration range yielding a coefficient of determination ($R^2$) ≥ 0.99 was selected as the linear detection range. The limit of detection (LOD) was determined by analyzing a dilution series of RNA or DNA targets subjected to *trans*-cleavage followed by fluorescence detection. The LOD was calculated using the formula LOD = 3σ/slope, where σ represents the standard deviation of three blank controls, and slope denotes the slope of the linear regression curve. This approach ensures robust and reproducible quantification of target analytes across the defined linear range.

### Statistics and reproducibility

No statistical method was used to predetermine sample size. No data were excluded from the analyses. The experiments were not randomized. The Investigators were not blinded to allocation during experiments and outcome assessment.

### Reporting summary

Further information on research design is available in the Nature Portfolio Reporting Summary linked to this article.

## Data availability

All data supporting this study are available within the article, Supplementary Files, and dedicated Source Data files. Structural coordinates referenced in this work are deposited under PDB ID: 5U34. Source data are provided with this paper.

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

## Acknowledgements
This work was supported by National Natural Science Foundation of China 22307150 (C.L.), Shenzhen Medical Research Fund A2303012 (C.L.), National Key Research and Development Program of China 2022YFE0201800 (C.Y.), Guangdong Basic and Applied Basic Research Foundation 2024A1515012319 (C.L.), Shenzhen Science and Technology Program JCYJ20230807110315032 (J.W.), JCYJ20240813150427036 (C.L.).

## Author contributions
J.W., X.Y. and Y.L. conceived and designed the research; J.W., X.Y., Y.L., W.L., X.Z. and W.Z. performed the experiments and data collection; C.Y. provided critical resources and technical support; J.W., X.Y., Y.L. and C.L. analysed and interpreted the data; C.L. supervised the project, secured funding, and provided conceptual guidance; J.W. and C.L. wrote the manuscript with input from all authors. All authors reviewed and approved the final version of the manuscript.

## Competing interests
The authors declare no competing interests.
