## [Transparent Peer Review File · Nature Communications]

Regulating Cleavage Activity and Enabling MicroRNA Detection with Split sgRNA in Cas12b

Corresponding Author: Professor Chaoxing Liu

Version 0:

Reviewer comments:

Reviewer #1

(Remarks to the Author)

Wang et al. report on a strategy of using split sgRNA to regulate the cleavage activity of the Cas12b enzyme, and demonstrate an application to the detection of microRNA. The authors evaluated two designs of split sgRNA and compared them with the conventional split format (tracrRNA and crRNA). Additionally, they incorporated various chemical modifications into the split module to broaden the potential applications of Cas12b-based assays. By replacing the split spacer with the target microRNA, they designed an assay and achieved direct detection of a target microRNA without the need for pre-amplification or transcription.

The core concept of the strategy used by Wang et al in this manuscript has been reported previously, including the manipulation of cleavage activity via split crRNA (Nat. Comm. 2024, 15, 8342) and the direct detection of microRNA using Cas12 family enzymes (J. Am. Chem. Soc., 2024, 39, 26657-26666, Nat. Comm. 2023, 14, 5409). The advances described in Wang et al appear to be incremental over the existing techniques.

One suggestion for the authors is to clearly articulate the novelty of this study in the context of existing literature. It appears that the authors applied a similar split crRNA approach as previously reported to a different Cas enzyme. Cas12a and Cas12b share many similarities, and the manuscript would benefit from a direct comparison of whether and how the split crRNA strategy functions differently—or yields unique advantages—when applied to Cas12b versus Cas12a. Such a comparison is essential to justify the innovation and relevance of this work.

The activation of a Cas system involves the binding of the crRNA (or sgRNA) with the Cas protein to form the ribonucleoprotein (RNP) and the interaction of the crRNA in the RNP with the target (activator). Manipulating the sgRNA has structural relevance. With the conventional crRNA, the pre-ordering and binding of the direct-repeat domain facilitate subsequent spacer binding and target recognition. Different split strategies are likely to induce distinct structural changes in the interactions between crRNA and the functional amino acids of the Cas protein. What might be structural implications of using split sgRNA from a structural biology perspective? Valuable structural insights could help explain the observed results from the comparison of the three split strategies.

Results in Figures 1, S3, and S4 show that the split structures, tracrDR+Spacer and tracrRNA+DR+Spacer, produce fewer cis-cleavage products but maintain a similar level of trans-cleavage activity, compared to the full-length sgRNA and the classic tracrRNA+crRNA format. Why? Both cis- cleavage and trans-cleavage activities of Cas12b share the same catalytic site. Previous studies on Cas12a have demonstrated that trans-cleavage was initiated only after the dissociation of cis-cleavage products. If fewer cis-cleavage products are observed in gel experiments, these results would suggest that the dissociation of cis-cleavage products might be limited. If this is the case, some mechanistic insight of how this limited dissociation results in similar trans-cleavage activity would be useful.

The chemical modifications on the split direct-repeat domain described in this study could also be applied to crRNA in the conventional crRNA-tracrRNA split format. Would direct chemical modifications on the direct-repeat domain affect the interactions of crRNA with magnesium ions or with the Rec lobe of Cas proteins? Providing insights into these potential effects is critical for evaluating the feasibility and mechanistic implications of the modifications.

The author claimed that the limit of detection for microRNA was at the fM level, which is almost three orders of magnitude lower than that for the detection of DNA targets. Although this trend is consistent with results of previous work (Nat. Commun. 2024, 15(1), 8342.), the microRNA activator was double strand DNA rather than single-stranded DNA activator. According to previous studies using the Cas12a system, ssDNA was more efficient than dsDNA in activating the trans-cleavage of Cas12a. Please clarify the apparent discrepancy.

Reviewer #2

(Remarks to the Author)

Wang et al. report a groundbreaking strategy for splitting the >100 nt sgRNA of Cas12b into a universal tracrDR (or tracrRNA + DR) and a replaceable ~20 nt Spacer. This approach is both ingenious and highly practical, addressing a critical challenge in the field: the precise regulation of Cas12b activity. Unlike Cas12a, Cas12b requires a longer sgRNA, making chemical modifications of the sgRNA and precise control of Cas12b activity more challenging. The authors provide robust experimental evidence demonstrating that their split sgRNA strategy (tracrDR + Spacer or tracrRNA + DR + Spacer) retains the functionality of full-length sgRNA. This innovation opens new avenues for the field, as it allows for targeting different sequences by simply replacing the Spacer while maintaining the universal components, offering significant potential for diagnostics and regulation.

Furthermore, the authors enhance the versatility of their strategy by introducing glyoxal or photo-cleavable (PC) linker modifications to the 16 nt DR, enabling temporal, thermal, or UV-light-controlled regulation of Cas12b activity. These modifications are highly innovative and provide valuable insights for future research. The use of a universal DR-PC-DNA construct to resolve the incompatibility between isothermal amplification and CRISPR reactions is particularly noteworthy. The successful application of this approach in clinical samples highlights its potential for trace target detection in clinical diagnostics.

The authors also demonstrate the replacement of the Spacer with miRNA of similar length in Cas12b split sgRNA, a strategy that aligns with recent findings by Chen et al. in Cas12a (Nat. Commun., 2024, 15, 8342). This work confirms that the miRNA replacement strategy is applicable to Cas12b, suggesting a potential universality across Cas proteins. Importantly, the split sgRNA strategy enhances Cas12b's single-base mismatch discrimination, a critical feature for precise diagnostics. The robust performance of this method in detecting miRNA panels in colorectal cancer and non-cancer cells, as well as in pre- and post-operative blood samples from colorectal cancer patients, underscores its potential for clinical applications, including colorectal cancer prognosis monitoring. These results are supported by extensive experimental data, making a compelling case for the translational relevance of this work.

Overall, this study represents a significant advancement in the CRISPR-Cas12b field. The authors provide a detailed and systematic exploration of their split sgRNA strategy, demonstrating its utility in precise regulation and medical applications. The work is well-executed, data-rich, and highly innovative. This study has the potential to inspire further advancements in CRISPR technology and its clinical applications. I recommend acceptance of this manuscript after addressing the following points:

1. The criteria for selecting linear ranges (e.g., Figure 2e) and the derivation of LOD values should be explicitly explained in Section 13 (Data Processing and Analysis) of the Methods.
2. In Figure S7, the authors test four methylation sites. Does the method's performance depend on the number of methylation sites? Clarifying this would enhance the generalizability of the approach.
3. In Figures 3b and S8a, the authors demonstrate glyoxal-mediated regulation of Cas12b activity but lack direct evidence of the interaction between glyoxal and the DR. Structural or biochemical evidence should be provided to support these claims.
4. Ensure consistent formatting throughout the manuscript, particularly in Figure 3 (e.g., "16nt" should be "16 nt," and "0min" should be "0 min").
5. Include the secondary structure of pre-miR-21 in the Supplementary Information and discuss the potential mechanisms underlying the selective detection of miR-21 over pre-miR-21. This would provide valuable insights for future studies, given the challenge of distinguishing them using traditional methods.
6. Many published studies use 60°C for Cas12b, while this manuscript primarily uses 48°C. The authors should provide a rationale for this choice, despite Cas12b having a broad working temperature range.
7. While the method appears robust and promising, the Discussion should include a section on its limitations to guide future in-depth investigations.
8. Consider moving the Methods section after the Discussion to emphasize the results and their implications.
9. Ensure all references are complete and formatted according to Nature Communications guidelines. For example, Ref. 38 is missing page numbers.

Version 1:

Reviewer comments:

Reviewer #1

(Remarks to the Author)

The authors have responded to reviewers' comments of the original manuscript, and have included additional results in Supporting Information. The new information and discussions on the dissociation constants are useful. The authors' attempts to provide some mechanistic understanding are also welcome additions to the revised manuscript. AlphaFold3 predictions and molecular dynamic simulations offered additional support, although the authors are suggested to soften their language by changing such wording as "validated" and "confirmed" to something like "provided supporting evidence".

The core concept of the strategy used by the authors in this manuscript has been reported previously. The authors argue that the Cas12b enzyme is quite different from the Cas12a enzyme system. Some of the relevant responses in the rebuttal letter should be incorporated into the manuscript (as concise key points) to delineate and justify the significance and relevance of

this work.

The authors should be more rigorous in presentation of the data. For example, in their new Figure S5, the authors show dissociation constant (k_d) values with excess significant figures. For example, they report 240.99 ± 10.86 . If the standard deviation (or error) is 10, how can they report with confidence any digit after the decimal point? By definition of "significant figures", only the last digit is uncertain. Reporting 241 ± 11 would be efficient, and reporting 240.99 ± 10.86 is erroneous. The authors are suggested to go through their manuscript and be rigorous with their reporting of values.

Reviewer #2

(Remarks to the Author)

The authors have given a detailed explanation to the questions raised by the reviewers and added relevant experiments. The revised manuscript has a more complete structure and better arguments. Therefore, the paper can be accepted for the journal.

Authors' Point-By-Point Response to the referees' comments

We sincerely thank all the referees for your time and efforts in thoroughly reading the manuscript and providing detailed comments as well as your expert insights. We believe our manuscript is now in a lot better shape due to your inputs, especially in terms of representing the data. We have addressed all the comments below with responses marked in blue and changes to the manuscript highlighted in yellow.

Reviewer #1 (Remarks to the Author):

Wang et al. report on a strategy of using split sgRNA to regulate the cleavage activity of the Cas12b enzyme, and demonstrate an application to the detection of microRNA. The authors evaluated two designs of split sgRNA and compared them with the conventional split format (tracrRNA and crRNA). Additionally, they incorporated various chemical modifications into the split module to broaden the potential applications of Cas12b-based assays. By replacing the split spacer with the target microRNA, they designed an assay and achieved direct detection of a target microRNA without the need for pre-amplification or transcription.

The core concept of the strategy used by Wang et al in this manuscript has been reported previously, including the manipulation of cleavage activity via split crRNA (Nat. Comm. 2024, 15, 8342) and the direct detection of microRNA using Cas12 family enzymes (J. Am. Chem. Soc., 2024, 39, 26657-26666, Nat. Comm. 2023, 14, 5409). The advances described in Wang et al appear to be incremental over the existing techniques.

One suggestion for the authors is to clearly articulate the novelty of this study in the context of existing literature. It appears that the authors applied a similar split crRNA approach as previously reported to a different Cas enzyme. Cas12a and Cas12b share many similarities, and the manuscript would benefit from a direct comparison of whether and how the split crRNA strategy functions differently—or yields unique advantages—when applied to Cas12b versus Cas12a. Such a comparison is essential to justify the innovation and relevance of this work.

Response:

We sincerely thank you for your thorough evaluation of our manuscript, your insightful suggestions, and your recognition and support of our research. We agree that Cas12b and Cas12a share similarities but also exhibit distinct differences. Compared to Cas12a, Cas12b demonstrates several superior properties, including enhanced thermostability, broader pH tolerance, and greater stability under physiological conditions. These attributes make Cas12b particularly well-suited for high-temperature applications.

Additionally, Cas12b's strong PAM preference (e.g., TTN) narrows its targeting range compared to certain Cas12a variants, potentially reducing off-target effects. Furthermore, Cas12b exhibits superior single-base resolution and target recognition specificity, enhancing its precision.

However, Cas12b also has notable limitations. For instance, while Cas12a requires only ~42 nt of crRNA for functionality and has seen significant advances in crRNA modifications—such as elongation, truncation, splitting, introduction of secondary structures, DNA embedding, and incorporation of modified nucleotides—Cas12b requires sgRNAs longer than 100 nt. This complicates chemical modifications of sgRNA for Cas12b's cleavage activity regulation and limits Cas12b's broader application. Consequently, the chemical modification of Cas12b's sgRNA and precise control of Cas12b's activity remain underexplored, highlighting a critical gap in the field.

In our revised manuscript, we have expanded the discussion to emphasize the significance of our split sgRNA strategy for Cas12b. Our work addresses this gap by demonstrating that split sgRNAs, despite their reduced affinity for Cas12b (as evidenced by increased dissociation constants), exhibit unique properties. Specifically, while *cis*-cleavage activity is diminished, the larger dissociation constant of the split sgRNA-Cas12b complex with target DNA facilitates easier target DNA dissociation than full-length sgRNA-Cas12b complex. Since *cis*- and *trans*-cleavage share the same catalytic site, this results in enhanced *trans*-cleavage activity. This hypothesis is supported by our observation that split sgRNAs yield fewer *cis*-cleavage products but maintain comparable *trans*-cleavage activity.

Moreover, we discovered that split sgRNA-assisted Cas12b exhibits superior single-base resolution compared to full-length sgRNA. Additionally, Cas12b's thermostability allows for the use of glyoxal-modified split sgRNAs, where elevated temperatures can rapidly restore activity—a feature not achievable with Cas12a. Furthermore, we have demonstrated that the affinity of the sgRNA-Cas12b complex for the target DNA is significantly lower than that of Cas12b for sgRNA (either in its split or full-length form). This finding suggests that integrating the target molecule (such as microRNA) as the Spacer portion of the split sgRNA, rather than detecting it as the target nucleic acid, can achieve superior detection sensitivity. Under your insightful guidance and inspiration, we have conducted a more in-depth exploration and analysis of the underlying mechanisms of the research findings, leading to the derivation of numerous intriguing and potentially valuable conclusions.

We are deeply grateful for your feedback and have incorporated additional experiments and discussions into the revised manuscript, as highlighted in the text, to underscore the innovation and significance of our work. These revisions further highlight the unique advantages of our split sgRNA strategy for Cas12b and its potential applications.

The activation of a Cas system involves the binding of the crRNA (or sgRNA) with the Cas protein to form the ribonucleoprotein (RNP) and the interaction of the crRNA in the RNP with the target (activator). Manipulating the sgRNA has structural relevance. With the conventional crRNA, the pre-ordering and binding of the direct-repeat domain facilitate subsequent spacer binding and target recognition. Different split strategies are likely to induce distinct structural changes in the interactions between crRNA and the functional amino acids of the Cas protein. What might be structural implications of using split sgRNA from a structural biology perspective? Valuable structural insights could help explain the observed results from the comparison of the three split strategies.

Response:

We sincerely appreciate your insightful comments and suggestions, which have significantly enhanced our understanding of the underlying mechanisms in this study. As you pointed out, the activation of the Cas system involves the formation of ribonucleoprotein (RNP) complexes through the binding of crRNA (or sgRNA) to Cas proteins, followed by the interaction between crRNA in the RNP and the target (activator). In our revised manuscript, we have supplemented the dissociation constants (K_d values) of Cas12b for full-length and three different split sgRNA, confirming that Cas12b exhibits stronger affinity for full-length sgRNA compared to split sgRNA.

Figure S5. Binding curves illustrating the response values as a function of Cas12b

concentration, determined using the Monolith Pico (NanoTemper) system, for the measurement of dissociation constants between Cas12b and (a) sgRNA, (b) tracrRNA+crRNA, (c) tracrDR+Spacer, and (d) tracrRNA+DR+Spacer. Error bars represent the standard deviation (n = 3).

Additionally, using AlphaFold3 predictions and molecular dynamics (MD) simulations, we validated the reliability of our simulations through root mean square deviation (RMSD) analysis. The average binding free energy analysis revealed that full-length sgRNA has the lowest binding free energy, which is corroborated by the Kd values obtained from the MicroScale Thermophoresis (MST) experiments. Root mean square fluctuation (RMSF) analysis was conducted to assess the stability of Cas12b in four systems. The REC1 region (residues 100-200) showed significant conformational changes. Specifically, in MD simulations, the REC1 region of the Cas12b-sgRNA complex stabilized within 30 ns, while the Cas12b-tracrRNA+crRNA complex stabilized after 90 ns. The Cas12b-tracrDR+Spacer complex stabilized at 60 ns, whereas the Cas12b-tracrRNA+DR+Spacer complex exhibited continuous conformational fluctuations throughout the 100 ns simulation. These results indicate that full-length sgRNA forms a more stable complex with Cas12b in the REC1 region compared to split forms.

Figure S7. Molecular dynamics simulation analysis of Cas12b interactions with full

length sgRNA or split sgRNA. (a) Time-dependent root-mean-square deviation (RMSD) of sgRNA, tracrRNA+crRNA, tracrDR+Spacer, and tracrRNA+DR+Spacer bound to Cas12b during the 0–100 ns simulation. All systems exhibit stable RMSD values after 30 ns, indicating reliable and meaningful simulations within the 100 ns timeframe. (b) Time-dependent binding free energy of sgRNA, tracrRNA+crRNA, tracrDR+Spacer, and tracrRNA+DR+Spacer bound to Cas12b during the 0–100 ns simulation. Full-length sgRNA demonstrates the lowest binding free energy, suggesting the strongest interaction with Cas12b. (c) Root-mean-square fluctuation (RMSF) analysis of Cas12b amino acid residues upon binding to full-length or split sgRNA. The y-axis represents RMSF (in nm), and the x-axis denotes the residue number. Higher RMSF values indicate greater conformational flexibility or reduced stability. Notably, residues 100–200 across all four systems exhibit elevated RMSF, suggesting increased flexibility or lower stability in this region.

Figure S8. Conformational snapshots of Cas12b interactions with full length sgRNA or split sgRNA during molecular dynamics simulations (0–100 ns, sampled every 10 ns). (a) sgRNA, (b) tracrRNA+crRNA, (c) tracrDR+Spacer, and (d) tracrRNA+DR+Spacer. Key structural features are highlighted: the REC1 domain of Cas12b is shown in red, the RuvC domain in green, and the Spacer region of sgRNA in cyan. Conformations before structural stabilization are marked with magenta circle, while blue circles indicate more stable conformations observed after stabilization. This visualization

provides insights into the dynamic behavior and structural transitions of the Cas12b-sgRNA complexes during the simulation.

The early stabilization of the REC1 region in the Cas12b-sgRNA complex (within 30 ns) facilitates the formation of an optimal target DNA binding pocket, accelerating the specific recognition process. This aligns with the enzyme-substrate induced fit theory and is consistent with MST results, which show that the interaction between full-length sgRNA and Cas12b has the lowest K_d . Cas12b utilizes a single RuvC nuclease domain for *cis*-cleavage of target dsDNA and *trans*-cleavage of non-specific ssDNA. We analyzed the Gibbs free energy contributions of the RuvC domain in the four systems. The tracrDR+Spacer split form exhibited Gibbs free energy contributions similar to full-length sgRNA, while the tracrRNA+DR+Spacer form interacted with more RuvC residues but with weaker interactions. Electrostatic and hydrogen bond analyses of RuvC and spacer regions indicated that the interactions between full-length and split sgRNAs with Cas12b involve different amino acid residues. This suggests that different split strategies may lead to significant structural changes in the interactions between sgRNA and functional amino acids of Cas12b.

Figure S9. Energy contribution profiles of Spacer regions to Cas12b RuvC domain residues in molecular dynamics simulations. The analysis includes sgRNA, tracrRNA+crRNA, tracrDR+Spacer, and tracrRNA+DR+Spacer. Notably, the split sgRNA exhibits significant variations in energy contributions to RuvC domain residues compared to full-length configurations, highlighting distinct interaction patterns.

Figure S10. Distribution of amino acid residues involved in hydrogen bonding and electrostatic interactions between the RuvC domain and Spacer region in full-length and split sgRNA complexes with Cas12b.

Furthermore, we measured the dissociation constants of the RNP complexes formed by full-length and split sgRNAs with target DNA. The full-length sgRNA and the classic split form (tracrRNA+crRNA) showed the lowest K_d values ($1.09 \pm 0.1 \mu\text{M}$ and $1.05 \pm 0.1 \mu\text{M}$, respectively). In contrast, our split forms tracrDR+Spacer ($4.13 \pm 0.72 \mu\text{M}$) and tracrRNA+DR+Spacer ($6.41 \pm 0.91 \mu\text{M}$) exhibited significantly higher K_d values, indicating weaker interactions with target DNA. As you mentioned, Cas12b's *cis*- and *trans*-cleavage activities share the same catalytic site. Previous studies on Cas12a have shown that *trans*-cleavage begins only after the dissociation of *cis*-cleavage products. Our gel experiments revealed that while our split strategies exhibited similar *trans*-cleavage activity, they produced fewer *cis*-cleavage products. This may be due to the higher dissociation constants of our split forms, leading to faster target dissociation and enhanced *trans*-cleavage activity under comparable *cis*-cleavage product conditions.

Figure S6. Binding curves depicting the response values in relation to the concentration of target nucleic acid for the either split or full-length sgRNA-Cas12b complexes, determined using the Monolith Pico (NanoTemper) system, to measure the dissociation constants of either split or full-length sgRNA-Cas12b complexes with target DNA. The curves are shown for (a) sgRNA, (b) tracrRNA+crRNA, (c) tracrDR+Spacer, and (d) tracrRNA+DR+Spacer. Error bars indicate the standard deviation (n = 3).

In our revised manuscript, we have added a new section titled "Mechanistic Insights into Split sgRNA Strategies on Cas12b Binding and Cleavage Activity," which includes relevant data and discussions. Your guidance has been invaluable, and our findings demonstrate that full-length sgRNA forms the most stable complex with Cas12b, exhibiting the highest binding affinity and optimal *cis*-cleavage activity. Although split forms show reduced binding affinity and *cis*-cleavage activity, their increased dissociation constants with target DNA result in enhanced trans-cleavage activity, maintaining trans-cleavage capabilities comparable to sgRNA. These structure-based findings and analyses provide deeper insights into how different split sgRNAs affect Cas12b cleavage activity and their potential applications in CRISPR-Cas12b technology. We are grateful for your comments, which have been extremely helpful.

Results in Figures 1, S3, and S4 show that the split structures, tracrDR+Spacer and tracrRNA+DR+Spacer, produce fewer *cis*-cleavage products but maintain a similar level of trans-cleavage activity, compared to the full-length sgRNA and the classic tracrRNA+crRNA format. Why? Both *cis*-cleavage and *trans*-cleavage activities of Cas12b share the same catalytic site. Previous studies on Cas12a have demonstrated that trans-cleavage was initiated only after the dissociation of *cis*-cleavage products. If fewer *cis*-cleavage products are observed in gel experiments, these results would suggest that the dissociation of *cis*-cleavage products might be limited. If this is the case, some mechanistic insight of how this limited dissociation results in similar trans-cleavage activity would be useful.

Response:

We sincerely appreciate your thorough review of our manuscript and your insightful comments. Your observation regarding our data prompted us to delve deeper into the underlying mechanisms. Inspired by your discussion, we have uncovered some intriguing potential mechanisms.

As you noted, compared to the full-length sgRNA and the classical tracrRNA+crRNA format, the split structures of tracrDR+Spacer and tracrRNA+DR+Spacer produce fewer *cis*-cleavage products but exhibit similar levels of trans-cleavage activity.

In the revised manuscript, we have examined the dissociation constants of Cas12b for the full-length sgRNA and the three split forms of sgRNA using microscale thermophoresis (MST) assays. Our findings indicate that Cas12b has a smaller dissociation constant for the full-length sgRNA compared to the split forms, suggesting a stronger affinity for the full-length sgRNA. Additionally, through AlphaFold3 predictions and MD simulations, we have revealed that the full-length sgRNA possesses a stronger binding free energy and a more stable binding structure with Cas12b. This implies that the full-length sgRNA is more likely to form a stable complex with Cas12b, thereby enhancing its *cis*-cleavage capability. Conversely, the split forms of sgRNA exhibit reduced affinity for Cas12b, resulting in fewer *cis*-cleavage products.

However, we also found that the dissociation constants of the sgRNA (full-length or split forms)-Cas12b complex for target nucleic acids differ. Our split forms-Cas12b complex have larger dissociation constants, indicating a stronger dissociation ability for target DNA. As you mentioned, Cas12b's *cis*-cleavage and trans-cleavage activities share the same catalytic site, with *trans*-cleavage commencing only after the dissociation of *cis*-cleavage products. The larger dissociation constants of the split sgRNA suggest a relatively stronger *trans*-cleavage ability. Consequently, we observed fewer *cis*-cleavage products but similar trans-cleavage activity.

Figure S5. Binding curves illustrating the response values as a function of Cas12b concentration, determined using the Monolith Pico (NanoTemper) system, for the measurement of dissociation constants between Cas12b and (a) sgRNA, (b) tracrRNA+crRNA, (c) tracrDR+Spacer, and (d) tracrRNA+DR+Spacer. Error bars represent the standard deviation (n = 3).

Figure S6. Binding curves depicting the response values in relation to the concentration of target nucleic acid for the either split or full-length sgRNA-Cas12b complexes, determined using the Monolith Pico (NanoTemper) system, to measure the dissociation constants of either split or full-length sgRNA-Cas12b complexes with target DNA. The curves are shown for (a) sgRNA, (b) tracrRNA+crRNA, (c) tracrDR+Spacer, and (d) tracrRNA+DR+Spacer. Error bars indicate the standard deviation (n = 3).

We are grateful for your suggestions and have incorporated the relevant data and conclusions into the revised manuscript.

The chemical modifications on the split direct-repeat domain described in this study could also be applied to crRNA in the conventional crRNA-tracrRNA split format. Would direct chemical modifications on the direct-repeat domain affect the interactions of crRNA with magnesium ions or with the Rec lobe of Cas proteins? Providing insights into these potential effects is critical for evaluating the feasibility and mechanistic

implications of the modifications.

Response:

We sincerely appreciate your valuable suggestions and insightful discussion. In our original manuscript, we demonstrated that the de-modification of glyoxal-modified full-length sgRNA resulted in reduced efficacy on Cas12b compared to our split sgRNA. Additionally, we highlighted that modification to the DR portion of the split sgRNA, which is a universal component capable of matching any target, eliminate the need for individual target-specific modifications. Furthermore, we noted that shorter sequences generally offer higher efficiency, lower cost, and greater yield in RNA synthesis, modification, and purification.

In response to your suggestions, we have included in the revised manuscript a structural analysis of the interaction between glyoxal-modified DR products and the Cas12b protein. Through AlphaFold3 predictions and MD simulations, we have revealed that the unmodified split sgRNA can form a stable complex with Cas12b. However, glyoxal modification of the DR significantly alters the binding mode of the sgRNA to Cas12b, preventing the Spacer from entering the functional cavity of Cas12b as it does in the normal split sgRNA. This alteration markedly inhibits the activity of Cas12b.

Figure S14. (d) Molecular dynamics simulations depicting the conformational interactions between Cas12b and the tracrRNA+DR+Spacer complex, compared with the tracrRNA+DR-glyoxal adduct+Spacer complex. Key structural features are highlighted, revealing that the DR-glyoxal adduct disrupts the normal binding of the tracrRNA+DR+Spacer complex to Cas12b, leading to altered conformational dynamics.

We are grateful for your input and have supplemented the revised manuscript with the relevant data.

The author claimed that the limit of detection for microRNA was at the fM level, which is almost three orders of magnitude lower than that for the detection of DNA targets. Although this trend is consistent with results of previous work (Nat. Commun. 2024, 15(1), 8342.), the microRNA activator was double strand DNA rather than single-stranded DNA activator. According to previous studies using the Cas12a system, ssDNA was more efficient than dsDNA in activating the trans-cleavage of Cas12a. Please clarify the apparent discrepancy.

Response:

We sincerely thank you for your insightful comments and constructive suggestions, which have significantly enhanced the depth and quality of our manuscript. Your observations prompted us to conduct a thorough investigation into the underlying mechanisms of the observed phenomenon.

As you correctly noted, our findings align with previous research (Nat. Commun. 2024, 15(1), 8342) demonstrating that microRNA serving as the spacer in sgRNA (or crRNA) exhibits a 2-3 orders of magnitude lower detection limit compared to conventional DNA as the target nucleic acid in sgRNA-Cas protein complexes. Building upon your comments, we have provided experimental evidence showing that Cas12b's dissociation constant for full-length or split sgRNA is significantly lower than that of the corresponding full-length or split sgRNA-Cas12b complex for target DNA. This data strongly suggests that Cas12b has a higher affinity for sgRNA.

Figure S5. Binding curves illustrating the response values as a function of Cas12b

concentration, determined using the Monolith Pico (NanoTemper) system, for the measurement of dissociation constants between Cas12b and (a) sgRNA, (b) tracrRNA+crRNA, (c) tracrDR+Spacer, and (d) tracrRNA+DR+Spacer. Error bars represent the standard deviation ($n = 3$).

Figure S6. Binding curves depicting the response values in relation to the concentration of target nucleic acid for the either split or full-length sgRNA-Cas12b complexes, determined using the Monolith Pico (NanoTemper) system, to measure the dissociation constants of either split or full-length sgRNA-Cas12b complexes with target DNA. The curves are shown for (a) sgRNA, (b) tracrRNA+crRNA, (c) tracrDR+Spacer, and (d) tracrRNA+DR+Spacer. Error bars indicate the standard deviation ($n = 3$).

Figure S33. (b) Real-time fluorescence intensity curves depicting the performance of the split sgRNA strategy (tracrDR) with a single-stranded DNA activator (miR-17 DNA activator, which is distinct from the double-stranded DNA activator used in Figure S24c.) at varying concentrations of miR-17 standards. (c) Data from panel (b) presented as a curve graph with linear regression analysis and limits of detection. The detection limits are comparable to those in Figure S24c, indicating that the use of either single-stranded or double-stranded DNA activators does not significantly affect the sensitivity of microRNA detection. (d) Comparative analysis of the detection performance of 500 nM tracrDR or tracrRNA+DR in complex with Cas12b and 500 nM double-stranded target DNA (as the Spacer activator) across a range of Spacer concentrations. The system achieves a detection limit as low as 10^2 fM. (e) Comparative analysis of the detection performance of 500 nM tracrDR+Spacer or tracrRNA+DR+Spacer in complex with Cas12b across a range of double-stranded target DNA concentrations. The system achieves a detection limit as low as 10^4 fM.

This mechanistic insight, which we have elaborated in the revised manuscript, explains why using the microRNA as the Spacer in split sgRNA results in a lower detection limit compared to using DNA as the target nucleic acid in the sgRNA-Cas protein complex. This understanding could potentially lead to the development of novel CRISPR-based

detection strategies in the future.

We have incorporated these findings and their implications into the revised manuscript, along with a comprehensive discussion, significantly enhancing the novelty and completeness of our work.

In conclusion, we are profoundly grateful for your thorough evaluation and constructive feedback, which have substantially improved the quality of our manuscript and allowed us to uncover several important mechanistic insights. Specifically, we have:

1. Compared the differences between Cas12b and Cas12a, highlighting the significance of our work as the first demonstration of the feasibility of splitting long Cas12b sgRNA into multiple fragments while maintaining its ability to guide Cas12b for both cis- and trans-cleavage.
2. Demonstrated that split sgRNA exhibits reduced affinity for Cas12b, leading to increased dissociation constants and weakened cis-cleavage activity. However, the increased dissociation constant of split sgRNA-Cas12b for target DNA results in enhanced trans-cleavage activity, making split sgRNA comparable to full-length sgRNA in terms of trans-cleavage while having relatively lower cis-cleavage activity. The present study also involves the other possible structural analyses of split and full-length sgRNAs interacting with Cas12b.
3. Investigated the potential structural mechanism by which glyoxal-labeled universal DR dynamically regulates CRISPR activity.
4. Established that microRNA can replace the spacer region (as part of split sgRNA) to guide Cas12b cleavage, with the detection sensitivity for microRNA being superior to that of traditional DNA as the target nucleic acid in sgRNA-Cas12b complexes. This enhanced sensitivity is attributed to Cas12b's higher affinity for sgRNA compared to the affinity of the sgRNA-Cas12b complex for target nucleic acids.

These comprehensive findings, made possible through your valuable input, have significantly strengthened the scientific rigor and impact of our manuscript.

Reviewer #2 (Remarks to the Author):

Wang et al. report a groundbreaking strategy for splitting the >100 nt sgRNA of Cas12b into a universal tracrDR (or tracrRNA + DR) and a replaceable ~20 nt Spacer. This approach is both ingenious and highly practical, addressing a critical challenge in the field: the precise regulation of Cas12b activity. Unlike Cas12a, Cas12b requires a longer sgRNA, making chemical modifications of the sgRNA and precise control of Cas12b activity more challenging. The authors provide robust experimental evidence

demonstrating that their split sgRNA strategy (tracrDR + Spacer or tracrRNA + DR + Spacer) retains the functionality of full-length sgRNA. This innovation opens new avenues for the field, as it allows for targeting different sequences by simply replacing the Spacer while maintaining the universal components, offering significant potential for diagnostics and regulation.

Furthermore, the authors enhance the versatility of their strategy by introducing glyoxal or photo-cleavable (PC) linker modifications to the 16 nt DR, enabling temporal, thermal, or UV-light-controlled regulation of Cas12b activity. These modifications are highly innovative and provide valuable insights for future research. The use of a universal DR-PC-DNA construct to resolve the incompatibility between isothermal amplification and CRISPR reactions is particularly noteworthy. The successful application of this approach in clinical samples highlights its potential for trace target detection in clinical diagnostics.

The authors also demonstrate the replacement of the Spacer with miRNA of similar length in Cas12b split sgRNA, a strategy that aligns with recent findings by Chen et al. in Cas12a (Nat. Commun., 2024, 15, 8342). This work confirms that the miRNA replacement strategy is applicable to Cas12b, suggesting a potential universality across Cas proteins. Importantly, the split sgRNA strategy enhances Cas12b's single-base mismatch discrimination, a critical feature for precise diagnostics. The robust performance of this method in detecting miRNA panels in colorectal cancer and non-cancer cells, as well as in pre- and post-operative blood samples from colorectal cancer patients, underscores its potential for clinical applications, including colorectal cancer prognosis monitoring. These results are supported by extensive experimental data, making a compelling case for the translational relevance of this work.

Overall, this study represents a significant advancement in the CRISPR-Cas12b field. The authors provide a detailed and systematic exploration of their split sgRNA strategy, demonstrating its utility in precise regulation and medical applications. The work is well-executed, data-rich, and highly innovative. This study has the potential to inspire further advancements in CRISPR technology and its clinical applications. I recommend acceptance of this manuscript after addressing the following points:

1. The criteria for selecting linear ranges (e.g., Figure 2e) and the derivation of LOD values should be explicitly explained in Section 13 (Data Processing and Analysis) of the Methods.

Response:

We sincerely appreciate your positive feedback on our research and are grateful for your constructive suggestions and insightful discussions.

Regarding the selection criteria for the linear range (as shown in Figure 2e) and the derivation of the LOD value, we have now provided a detailed explanation in Data Processing and Analysis of the Methods section. The concentration gradient of the standard target was prepared using a 2-fold serial dilution. The linear range was determined by plotting the fluorescence signal values against the corresponding concentrations of the standard target and performing linear regression analysis. We selected the concentration range with an R^2 value ≥ 0.99 for the final linear fitting. The limit of detection (LOD) was established by performing trans-cleavage assays with serially diluted RNA or DNA targets, followed by fluorescence detection. The LOD value was calculated using the formula $LOD = 3\sigma/\text{slope}$, where σ represents the standard deviation of three blank controls, and slope is the slope of the linear regression curve. We have incorporated these details into the revised manuscript as per your suggestion.

“For the determination of the linear detection range, a series of standard target concentrations were prepared using a 2-fold serial dilution. The linear range was identified by plotting the fluorescence signal values (y-axis) against the corresponding target concentrations (x-axis) and performing linear regression analysis. The concentration range yielding a coefficient of determination (R^2) ≥ 0.99 was selected as the linear detection range. The limit of detection (LOD) was determined by analyzing a dilution series of RNA or DNA targets subjected to trans-cleavage followed by fluorescence detection. The LOD was calculated using the formula $LOD = 3\sigma/\text{slope}$, where σ represents the standard deviation of three blank controls, and slope denotes the slope of the linear regression curve. This approach ensures robust and reproducible quantification of target analytes across the defined linear range.”

2. In Figure S7, the authors test four methylation sites. Does the method's performance depend on the number of methylation sites? Clarifying this would enhance the generalizability of the approach.

Response:

We are grateful for your thorough review of our manuscript. In our original submission,

Editorial Note: Supplementary Fig. 13a in this Peer Review File is adapted from Wang, J., Zhang, W., Li, W., Xie, Q., Zang, Z. and Liu, C. Enhancement of CRISPR-Cas12a system through universal circular RNA design. Cell Rep. Methods 5, 101076 (2025), licensed under CC-BY 4.0 (<https://creativecommons.org/licenses/by/4.0/>).

we indeed used a model chain of SEPT9 with four methylation sites, as hypermethylation of SEPT9, a diagnostic biomarker, rarely occurs at a single site. Your question and concerns have provided valuable insights for our study. While Cas12b already demonstrates strong single-base resolution capability, our split sgRNA design has further enhanced this feature (Figures 4e-g). In the revised manuscript, we have expanded our investigation by using SEPT9 model chains with 1-5 methylation sites. Our results demonstrate that Cas12b can effectively distinguish even a single methylation site (Figure S13). We greatly appreciate your suggestion, which has not only enhanced the novelty and systematic nature of our manuscript but also expanded its potential applications and enriched its content.

Figure S13. (a) Schematic representation of the tracrRNA+DR+Spacer-assisted CRISPR-Cas12b system for the detection of colorectal cancer methylation of Septin9. Methylated Septin9 undergoes selective oxidation of 5mC to generate DHU under the action of TAPS method, while unmethylated C remains unchanged.¹ Subsequently, under the action of RPA, the originally methylated sites are selectively converted to T.

Unmethylated Septin9 remains unchanged under the TAPS method. We can then detect it using the tracrRNA+DR+SEPT9 Spacer assisted CRISPR-Cas12b system. (b-f) Bar graph analysis of the detection of different ratios of methylated/unmethylated Septin9 using this method. (b) Target nucleic acid with one methylated site, (c) two, (d) three, (e) four, and (f) five methylated sites. Our research demonstrates that split sgRNA assisted Cas12b, with its exceptional single-base resolution in detecting target nucleic acids, can accurately identify differences even when the target sequence contains only a single methylation site. Error bars represent the standard deviation (n=3). Notes: ns, $P > 0.05$, *, $P \leq 0.01$, ***, $P \leq 0.001$, ****, $P \leq 0.0001$.

3. In Figures 3b and S8a, the authors demonstrate glyoxal-mediated regulation of Cas12b activity but lack direct evidence of the interaction between glyoxal and the DR. Structural or biochemical evidence should be provided to support these claims.

Response:

We sincerely appreciate your valuable suggestion regarding the need for direct evidence of the interaction between glyoxal and DR. In response to your comment, we have supplemented our revised manuscript with gel electrophoresis data that clearly demonstrates the formation of DR-glyoxal adducts. Additionally, we have included a time-course analysis (0-60 minutes at 60°C) showing the complete recovery of the DR structure from DR-glyoxal adduct after 40 minutes of heating. These new experimental results provide direct biochemical evidence supporting our claims about the glyoxal-mediated regulation of Cas12b activity, significantly strengthening the mechanistic foundation of our study.

Figure S14. Structural and functional characterization of DR-glyoxal adduct formation and its impact on Cas12b binding. (a) Schematic illustration of the molecular structure

of DR-glyoxal, formed through the reaction of RNA bases on DR with glyoxal. (b) Gel electrophoresis analysis of FAM-labeled DR following its reaction with glyoxal over a time course of 0–60 minutes. The results demonstrate successful formation of the DR-glyoxal adduct after 45 minutes of incubation.

Figure S15. Thermal recovery of RNA-glyoxal adduct and its impact on CRISPR-Cas12b trans-cleavage activity. (a) Gel electrophoresis analysis of the DR-glyoxal adduct heated at 60°C over a time course of 0–60 minutes, demonstrating the complete recovery of the DR structure after 40 minutes of heating

4. Ensure consistent formatting throughout the manuscript, particularly in Figure 3 (e.g., "16nt" should be "16 nt," and "0min" should be "0 min").

Response:

We thank you for pointing out the formatting inconsistencies in Figure 3. We have carefully reviewed the manuscript and made the necessary corrections, ensuring consistent formatting throughout. Specifically, we have standardized the presentation of units by adding spaces between numbers and their units (e.g., "16nt" to "16 nt" and "0min" to "0 min"). We appreciate your attention to detail, which has helped improve the overall quality and professionalism of our manuscript.

5. Include the secondary structure of pre-miR-21 in the Supplementary Information and discuss the potential mechanisms underlying the selective detection of miR-21 over pre-miR-21. This would provide valuable insights for future studies, given the challenge of distinguishing them using traditional methods.

Response:

We are grateful for your suggestion regarding the inclusion of pre-miR-21's secondary structure and the discussion of selective detection mechanisms. In response, we have added the secondary structure of pre-miR-21 to the Supplementary Information. Furthermore, we have expanded our discussion to include potential mechanisms underlying the selective detection of miR-21 over pre-miR-21. This addition not only

Editorial Note: Supplementary Fig. 23b in this Peer Review File is adapted from Wang, J., Zhang, W., Li, W., Xie, Q., Zang, Z. and Liu, C. Enhancement of CRISPR-Cas12a system through universal circular RNA design. Cell Rep. Methods 5, 101076 (2025), licensed under CC-BY 4.0 (<https://creativecommons.org/licenses/by/4.0/>).

addresses your specific concern but also provides valuable insights for future studies, particularly given the well-known challenges in distinguishing these molecules using traditional methods. We believe these enhancements will significantly contribute to the scientific value and impact of our work.

Figure S23. (a) Sequence and secondary structure of the miR-21 precursor, with the mature miR-21 sequence highlighted in red. (b) Schematic illustration of the split sgRNA strategy for detecting miR-21 and its precursor. In the miR-21 precursor, the mature miR-21 sequence is partially occluded by its complementary strand, preventing its binding to Cas12b and thereby inhibiting the activation of Cas12b's *trans*-cleavage activity. This design highlights the specificity of the split sgRNA approach in distinguishing between the mature miR-21 and its precursor. (c) Real-time quantitative fluorescence raw data validating the selective detection of different microRNAs using the split strategy (tracrDR) in conjunction with a DNA activator specific to miR-21. Notably, other microRNAs and the precursor of miR-21 do not produce fluorescence.

(d) Detection of miR-21 utilizing commercial test strips with the split strategy (tracrDR). The photograph displays the test strip results, with the T line indicating the presence of miR-21 and the C line serving as a control.

6. Many published studies use 60°C for Cas12b, while this manuscript primarily uses 48°C. The authors should provide a rationale for this choice, despite Cas12b having a broad working temperature range.

Response:

We appreciate your observation regarding our choice of 48°C as the primary working temperature for Cas12b. In response to your comment, we have included a detailed fluorescence analysis in the revised manuscript, which illustrates the *trans*-cleavage activity of Cas12b across a temperature range of 37–70°C. The data clearly show that both full-length and split sgRNA achieve peak fluorescence intensity at 48°C,

supporting our selection of this temperature as optimal for our experimental conditions. While Cas12b is indeed capable of functioning across a broad temperature range, our findings provide a strong rationale for using 48°C in our study. We have expanded the discussion to include this analysis, emphasizing that Cas12b has a broad applicable temperature range and can tolerate relatively higher temperatures.

Figure S33. (a) Fluorescence bar graph illustrating the trans-cleavage activity of Cas12b assisted by sgRNA, tracrRNA+crRNA, tracrDR+Spacer, and tracrRNA+DR+Spacer across a temperature range of 37–70°C. The results demonstrate that both full-length and split sgRNA exhibit peak fluorescence intensity at 48°C, confirming the robust temperature tolerance of Cas12b, including its stability at elevated temperatures.

7. While the method appears robust and promising, the Discussion should include a section on its limitations to guide future in-depth investigations.

Response:

Thank you for your suggestion to address the limitations of our method in the Discussion section. We have added a comprehensive analysis of the current constraints of our approach. Specifically, we acknowledge that while the split sgRNA strategy enhances trans-cleavage activity, it compromises cis-cleavage efficiency, which may limit its use in certain applications. Additionally, although the split strategy simplifies sgRNA design and enables direct microRNA detection, its sensitivity to target DNA remains suboptimal, particularly for trace clinical samples, necessitating integration with isothermal amplification methods. We also note that detection efficiency of SNP varies depending on the position of mutations within the spacer, requiring further optimization for clinical use. Finally, the increased experimental complexity and cost associated with small molecule modifications to the universal DR present a challenge that could be addressed through future commercialization of modified RNA components. These limitations provide valuable insights for guiding future research and development in this field.

“And our approaches also have limitations. The split sgRNA exhibits reduced affinity for Cas12b, which enhances *trans* cleavage activity but compromises *cis* cleavage efficiency, limiting its use in applications requiring robust *cis* cleavage. While the split strategy simplifies sgRNA design and allows direct detection of the microRNA as a spacer without reverse transcription or amplification, its sensitivity to target DNA remains suboptimal, particularly for trace clinical samples, necessitating integration with isothermal amplification methods. Although Cas12b demonstrates strong discrimination of single base mutations³ and our split strategy further improves this capability, detection efficiency varies depending on the position of the mutation within the Spacer, requiring additional optimization for clinical applications. Finally, small molecule modifications to the universal DR for precise regulation of Cas12b activity increase experimental complexity and cost, a challenge that could be mitigated by future commercialization of modified RNA components.”

8. Consider moving the Methods section after the Discussion to emphasize the results and their implications.

Response:

Thank you for your suggestions, we have moved the Methods section after the Discussion. This reorganization allows for greater emphasis on the results and their implications, improving the overall flow and readability of the manuscript. We believe this adjustment better aligns with the journal’s preferred structure and enhances the impact of our findings.

9. Ensure all references are complete and formatted according to Nature Communications guidelines. For example, Ref. 38 is missing page numbers.

Response:

We thank you for pointing out the incomplete reference formatting. We have carefully reviewed all references and ensured they are complete and formatted according to Nature Communications guidelines. Specifically, we have added the missing page numbers to Ref. 38 and verified the accuracy of all other references. This thorough review has improved the consistency and quality of our manuscript’s reference section.

Authors' Point-By-Point Response to the referees' comments

We sincerely thank all the referees for your time and efforts in thoroughly reading the manuscript and providing detailed comments as well as your expert insights. We believe our manuscript is now in a lot better shape due to your inputs. We have addressed all the comments below with responses marked in blue and changes to the revised manuscript highlighted in yellow.

Reviewer #1 (Remarks to the Author):

The authors have responded to reviewers' comments of the original manuscript, and have included additional results in Supporting Information. The new information and discussions on the dissociation constants are useful. The authors' attempts to provide some mechanistic understanding are also welcome additions to the revised manuscript. AlphaFold3 predictions and molecular dynamic simulations offered additional support, although the authors are suggested to soften their language by changing such wording as "validated" and "confirmed" to something like "provided supporting evidence".

Response:

We deeply appreciate your recognition of our manuscript improvements and your constructive feedback. Following your suggestion, we have carefully replaced terms like "validated" and "confirmed" with terms like "provided supporting evidence" throughout the manuscript, with all changes highlighted for easy identification. Your guidance has significantly enhanced the precision of our scientific language.

The core concept of the strategy used by the authors in this manuscript has been reported previously. The authors argue that the Cas12b enzyme is quite different from the Cas12a enzyme system. Some of the relevant responses in the rebuttal letter should be incorporated into the manuscript (as concise key points) to delineate and justify the significance and relevance of this work.

Response:

We are truly grateful for your insightful comments regarding the novelty of our work. In response, we have incorporated key points from our previous rebuttal letter into the manuscript's summary section (showing below), explicitly highlighting the distinct features of Cas12b compared to Cas12a and emphasizing our work's unique contributions. These additions strengthen both the clarity and impact of our presentation.

“In summary, Cas12b demonstrates several superior properties, including enhanced thermostability, broader pH tolerance, and greater stability under physiological conditions, making it particularly well-suited for high-temperature applications. Additionally, Cas12b’s strong PAM preference (e.g., TTN) narrows its targeting range compared to certain Cas12a variants, which may reduce off-target effects. Furthermore, Cas12b exhibits superior single-base resolution and target recognition specificity, further enhancing its precision. However, Cas12b also has several limitations compared to Cas12a. Notably, Cas12a functions efficiently with ~42 nt crRNA and has been optimized for crRNA modifications. In contrast, Cas12b requires significantly longer sgRNA (>100 nt). The increased length complicates the chemical modification of sgRNA, which hinders precise control of Cas12b’s cleavage activity and restricts its broader application. Here, we highlight the efficacy of the split sgRNA strategy in precisely regulating Cas12b functionality and expanding its application scope. Utilizing one to two universal components along with replaceable Spacer presents a modular design, promising to enhance nucleic acid detection across various clinical and research contexts. Future investigations will focus on further optimizing this platform for clinical evaluation and exploring its integration into broader diagnostic workflows.”

The authors should be more rigorous in presentation of the data. For example, in their new Figure S5, the authors show dissociation constant (kd) values with excess significant figures. For example, they report 240.99 +/- 10.86. If the standard deviation (or error) is 10, how can they report with confidence any digit after the decimal point? By definition of “significant figures”, only the last digit is uncertain. Reporting 241 +/- 11 would be efficient, and reporting 240.99 +/- 10.86 is erroneous. The authors are suggested to go through their manuscript and be rigorous with their reporting of values.

Response:

We sincerely apologize for the oversight in significant figure reporting and thank you for catching this important detail. We have rigorously corrected all numerical values in Figure S5, S6, and throughout the manuscript (e.g., changing 240.99 ± 10.86 to 241 ± 11), ensuring proper adherence to statistical reporting standards. Your attention to methodological rigor has greatly improved our data presentation.

Figure S5. Binding curves illustrating the response values as a function of Cas12b concentration, determined using the Monolith Pico (NanoTemper) system, for the measurement of dissociation constants between Cas12b and (a) sgRNA, (b) tracrRNA+crRNA, (c) tracrDR+Spacer, and (d) tracrRNA+DR+Spacer. Error bars represent the standard deviation ($n = 3$).

Figure S6. Binding curves depicting the response values in relation to the concentration of target nucleic acid for the either split or full-length sgRNA-Cas12b complexes, determined using the Monolith Pico (NanoTemper) system, to measure the dissociation constants of either split or full-length sgRNA-Cas12b complexes with target DNA. The curves are shown for (a) sgRNA, (b) tracrRNA+crRNA, (c) tracrDR+Spacer, and (d)

tracrRNA+DR+Spacer. Error bars indicate the standard deviation (n = 3).

Reviewer #2 (Remarks to the Author):

The authors have given a detailed explanation to the questions raised by the reviewers and added relevant experiments. The revised manuscript has a more complete structure and better arguments. Therefore, the paper can be accepted for the journal.

Response:

We are profoundly grateful for your positive evaluation and for recognizing the improvements in our revised manuscript. Your constructive feedback during the review process was instrumental in helping us strengthen both the experimental support and logical flow of our arguments. We particularly appreciate your endorsement for acceptance.